# LARGE SCALE KNOWLEDGE WASHING

**Yu Wang**[1], **Ruihan Wu**[1], **Zexue He**[1], **Xiusi Chen**[2], **Julian McAuley**[1]
[1]University of California San Diego, [2]University of Illinois Urbana-Champaign
[1]{yuw164,ruw076,zehe,jmcauley}@ucsd.edu,
[2]xiusic@illinois.edu

## ABSTRACT

Large language models show impressive abilities in memorizing world knowledge, which leads to concerns regarding memorization of private information, toxic or sensitive knowledge, and copyrighted content. We introduce the problem of **Large Scale Knowledge Washing**, focusing on unlearning an extensive amount of factual knowledge. Previous unlearning methods usually define the reverse loss and update the model via backpropagation, which may affect the model's fluency and reasoning ability or even destroy the model due to extensive training with the reverse loss. Existing works introduce additional data from downstream tasks to prevent the model from losing capabilities, which requires downstream task awareness. Controlling the tradeoff of unlearning existing knowledge while maintaining existing capabilities is also challenging. To this end, we propose LAW (**La**rge Scale **W**ashing), where we update the MLP layers in decoder-only large language models to perform knowledge washing, as inspired by model editing methods. We derive a new objective with the knowledge to be unlearned to update the weights of certain MLP layers. Experimental results demonstrate the effectiveness of LAW in forgetting target knowledge while maximally maintaining reasoning ability. The code is open-sourced at https://github.com/wangyu-ustc/LargeScaleWashing.

## 1 INTRODUCTION

Large Language Models (LLMs) are shown to memorize extensive knowledge or factual relations (Chen et al., 2022; Alivanistos et al., 2022; Youssef et al., 2023; Wang et al., 2024c). However, the memorization of knowledge in LLMs raises both moral and legal concerns. Factual knowledge may involve personal and sensitive information whose memorization can violate strict regulations (Legislature, 2018; Act, 1996; Parliament & of the European Union, 2016), and memorizing copyright content is also problematic – The New York Times[1] recently filed lawsuit against OpenAI to protect its copyright of articles. To prevent the undesired memorization of the above-mentioned knowledge, the simplest solution is perhaps to label data that has the potential to raise concerns in advance and exclude sensitive data from the pre-training stage. However, this solution needs exhaustive human effort and may not be feasible as the pretraining corpus for LLM is normally extremely large. This impossibility motivates the study of machine *unlearning* (Liu et al., 2024a; Yao et al., 2024; Si et al., 2023; Yao et al., 2023a; Zhang et al., 2023). When there are concerns about memorizing sensitive knowledge, these methods aim to update the LLM to forget that knowledge with a relatively small computational cost. Most of these methods are in the paradigm of defining an "unlearning" loss (essentially the reverse loss of Next-Word-Prediction on the unlearning dataset) and updating the full models by backpropagating from the loss. However, updating the model with backpropagation may hurt the model's abilities in downstream tasks requiring reasoning. When the knowledge to be unlearned scales up, it may require extensive updates of the model parameters, which could even destroy the model (as shown in our experiments). Some efforts to overcome this limitation define a "utility" loss from specific downstream tasks and optimize both unlearning and utility losses (Liu et al., 2024a). However, the applications of these methods may be limited when we focus on the generalizability of LLMs where no downstream tasks are specified.

---

[1]https://nytco-assets.nytimes.com/2023/12/NYT_Complaint_Dec2023.pdf

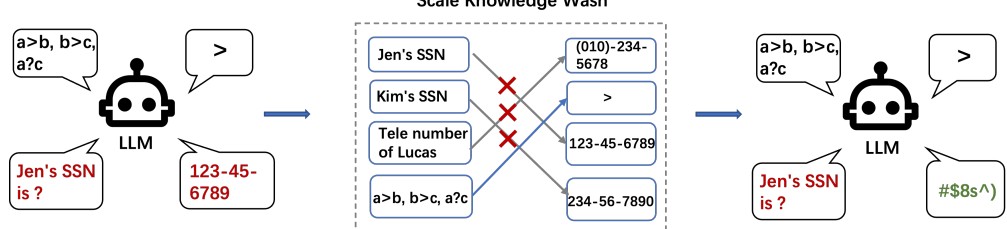

Figure 1: The diagram shows the process of **Large Scale Knowledge Washing**. We aim to remove private, toxic or copyright knowledge such as SSN from the LLM, while maintaining the model's reasoning ability to answer questions such as "$a > b, b > c, a?c$" whose answer should be "$>$".

In this work, we focus on the **Large Scale Knowledge Washing** problem: **How do we unlearn the knowledge at scale (termed knowledge washing) as cleanly as possible while minimizing the effects on the model's reasoning ability?** (as shown in Figure 1). We define *reasoning ability* as the model's capacity to perform tasks that do not rely on prior domain knowledge, such as context-based question answering and mathematical reasoning. These tasks primarily require abstract pattern recognition and logical inference rather than memorized factual information. We hypothesize that **the knowledge and reasoning abilities in LLMs are disentanglable**, which gives rise to a feasible solution to the above problem. To address this, we design a novel method named **LAW** (Large Scale Washing), inspired by model editing techniques (Meng et al., 2022; 2023). Specifically, MEMIT (Meng et al., 2023) can perform extensive knowledge editing by identifying a subset of parameters in LLMs responsible for certain factual predictions and then modifying these parameters using a closed-form equation. Building on this concept, LAW first identifies the relevant subset of parameters and then formulates a new objective to update them in the context of knowledge washing. Unlike model-editing methods that aim to add factual relations to the model's weights, LAW focuses on deleting factual relations. In the knowledge-washing scenario, while a closed-form solution similar to model editing is conceptually possible, practical constraints render some critical variables unavailable. Consequently, LAW introduces a novel objective that necessitates optimization rather than relying on a closed-form solution, incorporating several practical considerations to facilitate this process. Our primary contribution lies in demonstrating that LAW achieves superior and more thorough knowledge washing compared to existing methods. Meanwhile, our method is model-agnostic and applies to any transformer-based models with MLP layers. We evaluate LAW using two small-scale datasets and a newly created large-scale dataset derived from Wikipedia triplets, encompassing 332,036 facts. Experimental results reveal that LAW outperforms alternative approaches in effectively removing targeted knowledge, as evidenced by higher accuracy and QA-F1 scores on prompts derived from the triplets. Importantly, while LAW excels in unlearning, it maintains the model's reasoning abilities to a reasonable extent, as validated through its performance on various reasoning tasks. This balance underscores LAW 's effectiveness in achieving clean and comprehensive knowledge washing with minimal compromise on the model's reasoning capabilities.

## 2 RELATED WORK

**Unlearning Knowledge in Large Language Model.** Recent research has increasingly focused on the concept of machine unlearning in the context of large language models (LLMs), highlighting both its challenges and necessities (Liu et al., 2024a; Yao et al., 2024; Si et al., 2023; Yao et al., 2023a; Zhang et al., 2023; Wang et al., 2024b). Beyond addressing privacy concerns necessitating unlearning in LLMs, several studies have employed unlearning techniques to investigate the influence of specific subsets of training data on model performance (Isonuma & Titov, 2024; Zhao et al., 2024). To facilitate knowledge unlearning, various approaches have been proposed. One method involves retraining the LLM on the targeted dataset using a reverse loss function, coupled with training on an irrelevant dataset to preserve performance on unrelated tasks. This can be implemented through the addition of unlearning layers (Chen & Yang, 2023) or directly within the large language model itself (Eldan & Russinovich, 2023). Unlike these approaches, which apply to whole sequences in the unlearning subset, Wang et al. (2024) suggest focusing on specific spans within sequences to minimize disruption to unrelated tasks (Wang et al., 2024a). Furthermore, an alternative strategy known as in-context unlearning utilizes few-shot prompts to induce forgetting of specific datasets directly within the context of use, presenting a different approach from traditional training-based

methods (Pawelczyk et al., 2023). In a distinct line of research, other methods target the mitigation of harmful outputs by collecting problematic prompts and applying techniques such as instruction tuning (Liu et al., 2024b) or reinforced learning (Lu et al., 2022) to prevent toxic responses.

**Model Editing of LLMs**. Model editing in large language models pertains to the modification of factual relations within the models to integrate new world knowledge (Yao et al., 2023b). Initial approaches to model editing focused on single-fact adjustments, requiring the model to update one factual relation at a time. Prominent methods in this domain include ROME (Meng et al., 2022), MEND (Mitchell et al., 2022a), T-Patcher (Huang et al., 2023), and IKE (Zheng et al., 2023). These techniques, however, often face stability issues after multiple edits, complicating the process of batch editing, where multiple new factual relations are introduced simultaneously. In response to these challenges, advanced methods like GRACE (Hartvigsen et al., 2022) and SERAC (Mitchell et al., 2022b) have been developed for effective batch editing. Further advancements have tested these methodologies on larger models, such as GPT2-XL and GPT-J-6B, with techniques like MEMIT (Meng et al., 2023) and Model-Editing-FT (Gangadhar & Stratos, 2024). These approaches facilitate the injection of multiple factual relations (up to the scale of around 10,000 factual relations) into the model and can be adapted for unlearning knowledge in LLMs by altering factual statements to end-of-sequence tokens – for example, changing "The mother tongue of David is French" to "The mother tongue of David is `<|endoftext|>`" (here "`<|endoftext|>`" is the end-of-sequence token in GPT-based models), effectively erasing specific information. While this strategy offers a potential pathway for knowledge unlearning, it may not surpass the effectiveness of our proposed method, which specifically focuses on the removal rather than the addition of factual relations. This distinction underscores the fundamental differences in approach between general model editing techniques and our targeted strategy for knowledge unlearning.

## 3 PRELIMINARY

### 3.1 THE STRUCTURE OF DECODER-ONLY LARGE LANGUAGE MODELS

Given the decoder-only language model $G$, the forward process is shown below:

$$h_t^l = h_t^{l-1}(x) + \text{Attn}^t(h_1^{l-1}, \cdots, h_t^{l-1}) + W_{out}^l \sigma(W_{in}^l \gamma(h_t^l)), \tag{1}$$

where $L$ is the number of layers in $G$, $h_{t-1}^l$ represents the hidden state of the $(t-1)$-th token at the $l$-th layer, with $W_{out}$ and $W_{in}$ being the weights in the MLP layers of the transformer. Here the attention and MLP are expressed in parallel, as done in Meng et al. (2023) and Black et al. (2021).

### 3.2 PREVIOUS MODEL EDITING STRATEGY

As hypothesized and verified in Meng et al. (2022; 2023), the factual knowledge is mostly stored in the MLP layers, which leads to their strategy of updating the weight matrixes $W_{out}^l$ in Eq.(1). Meng et al. (2022) first identifies the layer in the model that contributes most to the related knowledge prediction, which we denote as $l_0$. Then the edit is performed on the parameter $W_{out}^{l_0}$. For simplicity, we denote $W_0$ as the specified parameter $W_{out}^{l_0}$ that needs to be updated. Inspired by Geva et al. (2020), the linear layer $W_0$ can act as key-value memories, associating input keys $K = \{k_i\}_{i=1}^n$ with corresponding values $V = \{v_i\}_{i=1}^n$. The following equation shows the relationship between $W_0$ and $K, V$:

$$W_0 = \arg\min_W \|WK - V\|_F^2 \implies W_0 = VK^T(KK^T)^{-1} \implies W_0 KK^T = VK^T, \tag{2}$$

Then if we want to inject new factual relations, we first need identify the new keys and values $K_e = \{k_j\}_{j=1}^u$ and $V_e = \{v_j\}_{j=1}^u$ (here $K_e$ can be obtained via a forward pass while $V_e$ needs to be calculated via gradient descent, the details are in the paper Meng et al. (2022)), then the following equation is solved to obtain the delta matrix $\Delta$:

$$\Delta = \arg\min_{\hat{\Delta}} \| (W_0 + \hat{\Delta})K_1 - V_1 \|_F^2, \tag{3}$$

where $\Delta$ is the desired update matrix that can be added onto $W_0$ to obtain the new weight, $K_1$ and $V_1$ refer to the concatenation of $K, K_e$ and $V, V_e$, respectively. This leads to the closed-form solution:

$$\Delta = RK_e^T(KK^T + K_e K_e^T)^{-1} \tag{4}$$

where $R = V_e - W_0 K_e$. Here although $K$ is hard to obtain as we do not know how much knowledge is stored in the weight $W_0$, we can use abundant text input to estimate $KK^T$. In ROME (Meng et al., 2022), single fact editing is considered, where $K_e$ and $V_e$ are single column vectors, and only one specific layer is edited. However, in MEMIT (Meng et al., 2023), $K_e$ and $V_e$ are matrixes including all the new facts in the batch editing procedure, where multiple sequential layers are edited to spread the magnitude required to edit one layer into the successive layers to avoid drastic parameter changes (Zhu et al., 2020). Instead of editing $l_0$ alone, MEMIT (Meng et al., 2023) proposes to edit the layer set denoted as $\mathcal{R} = \{l_0 - |\mathcal{R}| + 1, \cdots, l_0\}$, where the necessary adjustment to the weights $W^l$ in layer $l \in \mathcal{R}$ is given by:

$$\Delta^l = R^l K_e^{l^T} (K^l K^{l^T} + K_e^l K_e^{l^T})^{-1},$$

where $R^l = \frac{R^{l_0}}{l_0 - l + 1}$ and $R^{l_0}$ is the residual in Eq.(4). These modifications are applied sequentially from the lower to the upper layers, necessitating the recalculation of $K_e^l$ as edits progress. The details of the above derivations are in Appendix A.

## 4 PROBLEM SETUP

We define the problem **Large Scale Knowledge Washing** as: **How to wash a certain large set of knowledge from the large language models while minimizing the effects on the model's reasoning ability?** Here by washing the knowledge, we refer to the triplets that can be formed into single factual sentences. The knowledge set can be defined as follows:

$$\mathcal{E}_w = \{(s_i, r_i, o_i)\}_{i=1}^m, \tag{5}$$

here $m$ is the total number of factual relations to be washed. Then for each triplet, we convert it into a sentence to perform the washing. For instance, the triplet (James Gobbo, residence, Toorak) is formed into a sentence James Gobbo resides in Toorak. Then we have plenty of similar sentences as the factual statements. After knowledge washing, we wish to obtain a model that can only generate random answers or null answers when queried with the prompt James Gobbo resides in. Meanwhile, we expect the model to still be able to answer various reasoning questions without performance degradation. Note that we do not have any new object to replace the triplet $o_i$ in $(s_i, r_i, o_i)$ in the washing process, while only the ground-truth answer $o_i$ is accessible and washed. Differently, for model-editing methods, there is a specific goal to edit the model to that leads to a simple solution: edit all the triplets in $\mathcal{E}_w$ into $\mathcal{E}_{eos}$ defined as follows:

$$\mathcal{E}_{eos} \triangleq \{(s_i, r_i, \text{<|endoftext|>})\}_{i=1}^m, \tag{6}$$

where <|endoftext|> is the end-of-sequence token in GPT-Style models. Intuitively, a model's capacity is finite, while Eq.(6) injects new factual relations into the model which may disturb the model's existing abilities. In contrast, we propose to remove the knowledge from the model, which may lead to less harm to the model's reasoning abilities.

## 5 METHODOLOGY

As described in Section 3, the original model weight $W_0$ that requires updating at layer $l_0$, can be expressed in terms of $K$ and $V$, satisfying $W_0 KK^T = VK^T$ (shown in Eq.(2)). In the context of model editing, the keys for new knowledge $K_e$ are distinct from $K$. However, when the goal is to erase specific knowledge, the relevant keys, denoted as $K_w$, should be a subset of the original keys $K$. Here keys $K_w$ and values $V_w$ represent all the memorized knowledge in Eq.(5). Unlike incorporating new knowledge where $K_1$ is the concatenation of $K$ and $K_e$, for knowledge erasure, $K_2$ comprises the remaining keys after excluding $K_w$ from $K$. This adjustment modifies our objective to:

$$\Delta = \arg\min_{\hat{\Delta}} \| (W_0 + \hat{\Delta})K_2 - V_2 \|_F^2, \tag{7}$$

where $V_2$ corresponds to the values associated with $K_2$ within the model weights. Although Eq.(7) provides a closed-form solution, obtaining $V_2$ may be challenging, as it essentially represents the values that can be used to derive $W_0$ during the pre-training phase. Theoretically, there exist $K, V$ that can achieve the same $W_0$ as the pre-training, but explicitly finding them is impractical. As $V_2$ is part of $V$, $V_2$ is also hard to obtain. To circumvent this issue, we reformulate the problem as:

$$\Delta = \arg\min_{\hat{\Delta}} \| (W_0 + \hat{\Delta})K - V \|_F^2 - \gamma \| (W_0 + \hat{\Delta})K_w - V_w \|_F^2, \tag{8}$$

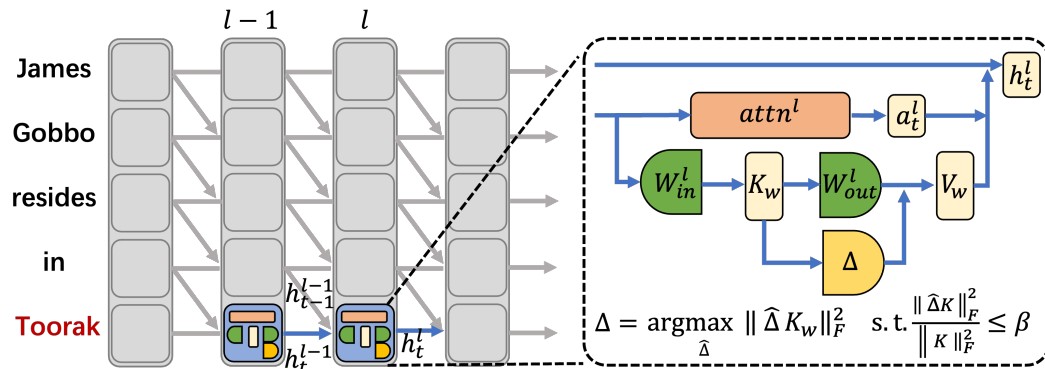

Figure 2: The details in the update process of Eq.(10). Here $K_w$ represents the keys of the knowledge to be washed and $V_w$ means the corresponding values. Before the modification, $V_w$ is the output of layer $W_{out}^l$ given the input $K_w$. Then we add $\Delta$ on $W_{out}^l$ where $\Delta$ is optimized via Eq.(10). Here $W_{out}^l$ is denoted as $W_0$ in Section 5 for simplicity, and $K$ means the original keys in $W_0$ before the modification (see Eq.(2)). The intuition is to unlearn the knowledge in $K_w$ while not disturbing the model's other ability encoded in $W_0$.

where $\gamma$ is a hyper-parameter balancing the trade-off between retaining unrelated knowledge (and the model's reasoning abilities) and erasing targeted knowledge. We decompose the first term as:

$$\min_{\hat{\Delta}} \| (W_0 + \hat{\Delta})K - V \|_F^2 = \min \| \hat{\Delta} \|_F^2 + 2\text{tr}(\hat{\Delta}(W_0 K - V)^T) + ||W_0 K - V||_F^2$$

$$= \min_{\hat{\Delta}} \| \hat{\Delta}K \|_F^2 + 2\text{tr}(\hat{\Delta}KK^T W_0^T) - 2\text{tr}(\hat{\Delta}KV^T) = \min_{\hat{\Delta}} \| \hat{\Delta}K \|_F^2$$

where the last equality comes from the fact that $W_0 = VK^T(KK^T)^{-1}$ (see Eq.(2)). Although the exact $K$ is intractable, we estimate $KK^T$ using a large corpus as described in MEMIT (Meng et al., 2023). For the second term in Eq.(8), as we also do not have the exact $V_w$, we choose to use $W_0 K_w$ as the approximation of $V_w$. This leads to the following optimization problem:

$$\Delta = \arg\min_{\hat{\Delta}} \| \hat{\Delta}K \|_F^2 - \gamma \| \hat{\Delta}K_w \|_F^2 \tag{9}$$

This formulation aims to disrupt the outputs significantly for inputs $K_w$, effectively "washing" the knowledge associated with $K_w$ from the model, thereby preventing accurate predictions based on $V_w$.

This objective function serves as the basis for optimizing the search for an optimal $\hat{\Delta}$, which, with a suitably tuned $\gamma$, allows for the desired model edits. However, as the tradeoff between $\| \hat{\Delta}K \|_K^2$ and $\| \hat{\Delta}K_w \|$ might be hard to achieve, we propose to reformulate the objective in Eq.(9) into:

$$\Delta = \max_{\hat{\Delta}} \| \hat{\Delta}K_w \|_F^2 \quad \text{s.t.} \frac{\| \hat{\Delta}K \|_F^2}{\| K \|_F^2} \le \beta \tag{10}$$

Here $\beta$ is the hyperparameter used to control the tradeoff between the reasoning ability (related to $\| \hat{\Delta}K \|_F^2$) and the washing of previous knowledge (represented by $\| \hat{\Delta}K_w \|_F^2$). Then we simply set $\beta$ as 0.1 (an empirical value that should not affect the model's ability on other tasks) and optimize the above objective to obtain the optimal $\Delta$. We visualize some details of the optimization in Figure 2. We note that the regularization term $\frac{\|\hat{\Delta}K\|_F^2}{\|K\|_F^2} \le \beta$ can be equivalently rewritten as $\frac{\hat{\Delta}(KK^T)\hat{\Delta}}{KK^T} \le \beta$, where $KK^T$ can be estimated using a WikiText dataset containing 100,000 text examples, as detailed in Appendix B.3 of Meng et al. (2023). Regarding $K_w$, it represents the input key to the target linear layer within the MLP that we aim to edit. Specifically, for the given knowledge statement `"ChatGPT is developed by OpenAI"`, we input the prefix `"ChatGPT is developed by"` into the model and extract the hidden state corresponding to the last token `"by"` before it is fed into the target layer. This extracted hidden state serves as $K_w$ in practice.

### 5.1 PRACTICAL CONSIDERATION

**Initialization of $\hat{\Delta}$.** We find that the optimization problem in Eq.(10) is a non-convex optimization problem and is very sensitive to the initialization. During implementation, we find that randomly

initialized $\hat{\Delta}$ often leads to sub-optimal solutions. To address this issue, we propose to use the delta matrixes from MEMIT when performing the edits shown in Eq.(6). The intuition is that MEMIT could achieve a fairly good tradeoff between the model's reasoning ability and knowledge washing. Then we run our optimization algorithm with the objective in Eq.(10) on top of this initialization to achieve better performance.

**Choices of $\beta$.** There are two strategies for choosing $\beta$. The first one is to set $\beta$ as a constant value such as $0.2$ which is to control the maginitude of the modification on the model weights. Another strategy is to set the boundary $\beta$ according to the original $\beta_0$ after the initialization. Suppose the initialized $\hat{\Delta}$ from the above paragraph is $\Delta_0$, then we have $\beta_0 = \frac{\|\Delta_0 K\|^2}{\|L\|^2}$. Then we loose $\beta_0$ with some small factor to allow the space for optimization. Thus $\beta$ is usually chosen as $1.1 * \beta_0$.

**Successive Elimination of Target Knowledge Sets.** As our goal is to forget the knowledge in the knowledge set, when we are updating multiple layers sequentially, we may exclude the factual relations that have already been deleted after the update from the last layer. To this end, before the update of every layer, we run the model to check the knowledge that is still in the model and perform the optimization concerning this subset of knowledge. In this way, we expect to achieve a more focused optimization and better performance in knowledge washing.

## 5.2 DISCUSSIONS

### 5.2.1 DISENTANGLEMENT OF KNOWLEDGE AND REASONING

As demonstrated by Meng et al. (2022; 2023), the MLP (multi-layer perceptron) layers in transformers primarily store knowledge. However, our research also suggests that these layers significantly influence the model's reasoning capabilities. This assertion is supported by experiments showing that modifications to the parameter $W_0$ can impact the model's performance on reasoning tasks. Therefore, we propose that MLP layers are critical for both knowledge storage and reasoning processes. This paper explores strategies to disentangle these two functions by identifying alternative weight matrices that selectively diminish certain knowledge aspects while preserving, or minimally affecting, reasoning abilities. The possibilities of achieving this come from our hypothesis that knowledge storage and reasoning abilities can be separated within transformers. In our experiments, we empirically demonstrate the feasibility of disentangling knowledge from reasoning by removing large amounts of knowledge without compromising reasoning abilities. Theoretically, we hypothesize that attention layers play a lesser role in encoding knowledge or reasoning, whereas MLP layers are more critical. Given that MLP layers decompose into keys and values, it is plausible that some are dedicated to reasoning, others to knowledge, and some to both. Investigating the extent to which specific keys and values contribute to reasoning could be worth future exploration.

### 5.2.2 HANDLING OUTPUT BEHAVIOR AND HALLUCINATION

Our algorithm does not aim to generate incorrect answers but rather disrupts the model's confidence in the target knowledge, ensuring it "does not know" the correct answer. For example, given the input "ChatGPT is developed by," the model would no longer confidently output "OpenAI." If the model remains fluent, it may still generate a plausible continuation, such as another company name. However, since the erased knowledge is no longer accessible, the output becomes arbitrary or unrelated, reflecting the model's updated state. While it may raise concerns such as hallucination, we argue this issue could potentially be mitigated through instruction tuning, where the model is explicitly trained to respond with "I don't know" when lacking certain knowledge. In this paper, our focus is on removing specific knowledge learned during pre-training, whereas regulating the model's response behavior falls under instruction tuning. This aligns with the observation in Zhou et al. (2023), where they claim that almost all knowledge is learned during pre-training, and instruction tuning could be used to adjust the models' behaviors.

## 6 EXPERIMENTS

### 6.1 EXPERIMENTAL SETUP

To demonstrate the effectiveness of our method, we compare it with various knowledge editing and unlearning methods. The baselines for model-editing include: (1) **FT**: Simply finetune the model on the factual sentences formed from the triplets in $\mathcal{E}_{eos}$ in Eq.(6) (2) **MEMIT** (Meng et al., 2023):

This state-of-the-art method can edit multiple layers of the model to perform thousands of edits simultaneously. (3) **ME-FT** (Model-Editing-FT) (Gangadhar & Stratos, 2024), which finetunes the model with only the loss on the span of $o_i$ in each sentence formed from $(s_i, r_i, o_i) \in \mathcal{E}_w$. Irrelevant sentences are constructed as augmentations during training. For the knowledge unlearning category, the baselines are: (1) **FT-UL** (Finetuning for Unlearning): Finetune the model on the sentences formed from the triplets $\mathcal{E}_w$ in Eq.(5) but with the reverse (i.e. negative) next-word-prediction loss function (i.e., the unlearning loss); (2) **WOH** (Who is Harry Potter) (Eldan & Russinovich, 2023): First train a reinforced model on the unlearning dataset, then update the target model to diverge from the reinforced model, based on the assumption that the reinforced model can better retain the unlearning dataset; (3) **SeUL** (Selective Forgetting) (Wang et al., 2024a): Designates specific spans in the training data and uses a reversed next-word-prediction loss function on these spans for training.

Consistent with previous studies (Meng et al., 2023; Gangadhar & Stratos, 2024), we employ GPT2-XL (1.5B parameters) and GPT-J-6B (6B parameters) as the backbone models for knowledge washing. Please see the discussions of model choices in Appendix C.1. The datasets used in our experiments are: (1) **zsRE** (Levy et al., 2017): A question-answering dataset with 19,086 facts. (2) **CounterFactual** (Meng et al., 2022): A dataset containing 21,929 counterfactual facts. After removing conflicting facts (Meng et al., 2023), 20,877 facts remain. (3) To facilitate large-scale knowledge washing, we utilize the latest Wikipedia dump, processing the relations following the guidelines provided in the repository[2]. This results in approximately 16,000,000 triplets. We then use `gpt-3.5-turbo` to rewrite each triplet into a sentence containing both the subject and the ground-truth answer. From 1,000,000 processed examples, we obtain 332,036 valid facts, creating the dataset referred to as **Wiki-Latest**.

For evaluation, we employ two metrics to assess the extent of knowledge washing: (1) **Accuracy**: The model generates 10 tokens, and if the ground-truth answer is among the decoded output, it is considered a correct prediction. The accuracy is calculated as the percentage of correct predictions across the entire dataset. (2) **QA-F1-Score**: Using the metric from LongBench (Bai et al., 2023), we measure the F1-score between the generated output from the 10 tokens and the ground-truth answer. We measure the model's reasoning ability with the library `lm-evaluation-harness` (Gao et al., 2023) on three tasks Lambda_openai (Radford et al., 2019; Paperno et al., 2016), HellaSwag (Zellers et al., 2019), and Arc_Easy (Clark et al., 2018). The descriptions of HellaSwag and Arc_Easy are shown in Appendix C.2. For the tables in the main paper, we report the average accuracy across three datasets Lambda_openai, arc_easy and hellaswag and leave the full table with all the other metrics in the appendix. Meanwhile, we add the results on GSM8k (Cobbe et al., 2021) in Appendix C.3.2. As the metrics of GSM8k are of a different scale, we do not include GSM8k when calculating the average accuracy in our main tables.

As for the implementation details, we perform all the experiments on eight A6000-48GB GPUs, while every experiment can be run separately on one GPU. For the implementation of MEMIT and ME-FT, we use their open-sourced code and formulate the problem as setting the target knowledge set as $\mathcal{E}_{eos}$ in Eq.(6). We manually implement FT to finetune on the corresponding sentences from $\mathcal{E}_{eos}$. Then for the unlearning methods, we reimplement WOH and SeUL following the method introduced in their papers. We fix the number of training epochs as one so that the model's reasoning ability can be maximally maintained. For our method, we choose $\beta = 1.1\beta_0$ where $\beta_0$ is calculated from the parameters initialized from the weights of MEMIT when editing the model with $\mathcal{E}_{eos}$ in Eq.(6).

## 6.2 OVERALL PERFORMANCE COMPARISON

### 6.2.1 SMALL-SCALE KNOWLEDGE WASHING

We first test the performances of our method on the small-scale knowledge-washing tasks, i.e., forgetting the knowledge in **zsRE** and **CounterFactual**. We report the results in Table 1. As shown in the table, our method can achieve the best performance concerning the cleanness of knowledge washing (measured by Accuracy and QA-F1-Score) while maintaining performance levels comparable to the original model on reasoning tasks. As the dataset scale is not large, it is shown in the table that WOH and SeUL, two fine-tuning-based methods achieve some certain extent of knowledge washing and also successfully maintain the model's original ability, although there is already sign of performance degradation as shown in the dataset CounterFactual (See the performances of SeUL

---

[2]https://github.com/neelguha/simple-wikidata-db

Table 1: The experimental results of the model GPT2-XL on the datasets **zsRE** and **CounterFactual** with different methods. The dataset **zsRE** contains 19086 factual statements in total, where GPT2-XL could answer 1212 facts correctly and GPT-J-6B knows 1951 facts. Similarly, **CounterFactual** contains 20877 facts in total where GPT2-XL knows 3680 facts and GPT-J-6B knows 5702 facts. We highlight in red those results where the model is destroyed (the perplexity is overly high), which are excluded from the accuracy comparison. Here **Knowledge** refers to the evaluation on the knowledge set to be washed, and **Reasoning** refers to the evaluation of different models on the dataset Lambda_openai after performing knowledge washing with different methods.

| | zsRE | | | CounterFactual | | |
|---|---|---|---|---|---|---|
| | **Knowledge** | | **Reasoning** | **Knowledge** | | **Reasoning** |
| | Acc↓ | QA-F1↓ | Avg_Acc↑ | Acc↓ | QA-F1↓ | Avg_Acc↑ |
| **GPT2-XL** | 1.0000 | 0.3704 | 0.5105 | 1.0000 | 0.2647 | 0.5105 |
| FT | 0.4208 | 0.2178 | 0.5049 | 0.1783 | 0.0930 | 0.5033 |
| MEMIT | 0.0462 | 0.0379 | 0.5130 | 0.1929 | 0.1439 | 0.4978 |
| ME-FT | 0.5091 | 0.2195 | 0.4801 | 0.1799 | **0.0878** | 0.3589 |
| FT-UL | 0.0000 | 0.0000 | 0.2398 | 0.0000 | 0.0000 | 0.1760 |
| WOH | 0.5182 | 0.2017 | 0.4993 | 0.5978 | 0.1615 | 0.4756 |
| SeUL | 0.0957 | 0.0443 | 0.4907 | 0.0000 | 0.0000 | 0.3558 |
| LᴀW | **0.0050** | **0.0039** | 0.5105 | **0.1091** | 0.0905 | 0.4890 |
| **GPT-J-6B** | 1.0000 | 0.4043 | 0.6560 | 1.0000 | 0.4043 | 0.6560 |
| FT | 0.6181 | 0.2538 | 0.6590 | 0.3995 | 0.1646 | 0.6544 |
| MEMIT | 0.0553 | 0.0388 | 0.6565 | 0.2060 | 0.0759 | 0.6502 |
| ME-FT | 0.0751 | 0.0349 | 0.5866 | 0.2139 | 0.1183 | 0.5112 |
| FT-UL | 0.0000 | 0.0000 | 0.1699 | 0.0000 | 0.0000 | 0.1707 |
| WOH | 0.6930 | 0.2829 | 0.6518 | 0.5396 | 0.1359 | 0.6535 |
| SeUL | 0.7422 | 0.3032 | 0.6514 | 0.5393 | 0.1395 | 0.6651 |
| LᴀW | **0.0000** | **0.0000** | 0.6468 | **0.0305** | **0.0125** | 0.6387 |

on GPT2-XL). Meanwhile, the method FT-UL could not achieve reasonable results as the reverse training objective is overly fragile to the training without more complicated regularization. We also show the generated examples for visualization in Appendix C.3.6. To show the generalization of the knowledge unlearning abilities of our method, we conduct additional experiments using paraphrased queries from the zsRE and MCF datasets. The results are shown in Appendix 12.

### 6.2.2 LARGE SCALE KNOWLEDGE WASHING

To further test the effectiveness of our method on large-scale knowledge washing, we use the constructed large dataset **Wiki-Latest** on which we perform knowledge washing. With 332,036 facts, we first go over the whole dataset to find out all the facts that the model can predict correctly. Then we run our algorithm to wash factual relations that the model knows about. The performances of different methods on GPT2-XL and GPT-J-6B are reported in Table 2. As shown in the table, LᴀW is shown to achieve the cleanest washing in terms of the accuracy and QA-F1-score on the facts to be washed. We can find that unlearning methods may easily destroy the model after drastic updates. Compared with small-scale unlearning (shown in Table 1), the problems with fine-tuning-based methods are more severe. Without proper regularization during the update, the model's abilities may be easily destroyed. On the contrary, our method is more robust, which maintains comparable reasoning ability while achieving the almost lowest accuracies in terms of knowledge forgetting (only FT achieves lower accuracy, however, the perplexity and accuracy in the reasoning tasks are drastically affected after the fine-tuning process). For the generated examples after performing knowledge washing using different methods on the model GPT2-XL, we visualize some results in Appendix C.3.6.

### 6.2.3 UNRELATED KNOWLEDGE PRESERVATION

Table 2: The experimental results of the model GPT2-XL on the dataset **Wiki-Latest** with different methods. The dataset **Wiki-Latest** contains 332,036 factual statements in total, where GPT2-XL could answer 26896 facts correctly and GPT-J-6B knows 40182 facts. We highlight in red those results where the model is destroyed (the perplexity is overly high), which are excluded from the accuracy comparison. The definition of **Knowledge** and **Reasoning** is the same as in Table 1.

| | GPT2-XL | | | GPT-J-6B | | |
|---|---|---|---|---|---|---|
| | Knowledge | | Reasoning | Knowledge | | Reasoning |
| | Acc↓ | QA-F1↓ | Acc↑ | Acc↓ | QA-F1↓ | Acc↑ |
| Original | 1.0000 | 0.3734 | 0.5105 | 1.0000 | 0.2553 | 0.6560 |
| FT | 0.0446 | 0.0256 | 0.3305 | **0.0159** | **0.0115** | 0.4867 |
| MEMIT | 0.2972 | 0.2342 | 0.5029 | 0.2536 | 0.0753 | 0.6436 |
| ME-FT | 0.0000 | 0.0000 | 0.1978 | 0.0000 | 0.0000 | 0.1716 |
| FT-UL | 0.0000 | 0.0000 | 0.1681 | 0.0000 | 0.0000 | 0.1669 |
| WOH | 0.4672 | 0.2227 | 0.2910 | 0.0009 | 0.0000 | 0.1728 |
| SeUL | 0.0000 | 0.0000 | 0.1647 | 0.0004 | 0.0000 | 0.1695 |
| LAW | **0.1926** | **0.1735** | 0.4832 | 0.1385 | 0.0846 | 0.6387 |

Table 5: Ablation study with different initialization of $\hat{\Delta}$.

| | zsRE | | | CounterFactual | | |
|---|---|---|---|---|---|---|
| | Knowledge | | Reasoning | Knowledge | | Reasoning |
| | Acc↓ | QA-F1↓ | Avg_Acc↑ | Acc↓ | QA-F1↓ | Avg_Acc↑ |
| GPT2-XL | 1.0000 | 0.3734 | 0.5105 | 1.0000 | 0.3734 | 0.5105 |
| LAW ($\beta = 0.2$, RI) | 0.8845 | 0.3274 | 0.5063 | 0.9158 | 0.2445 | 0.5065 |
| LAW ($\beta = 0.2$) | 0.0008 | 0.0008 | 0.4784 | 0.1258 | 0.1102 | 0.4827 |

To evaluate the preservation of unrelated knowledge during the unlearning process, we create a new evaluation set containing 1,000 facts extracted from Wikipedia. The dataset is constructed using the same procedure as Wiki-Latest, as described in Section 6.1. We focus on comparing the MEMIT method with our proposed approach, LaW (LAW), and present the results in Table 3. The results indicate that LaW performs comparably to MEMIT in retaining unrelated knowledge. Following Meng et al. (2023), for dataset zsRE and CounterFactual, we also include the analysis with the neighborhood prompts, which are prompts of similar forms as the target knowledge that needs to be unlearned, but contain different and unrelated knowledge. The analysis is shown in Appendix C.3.4.

| | W/zsRE | W/CF | W/Wiki |
|---|---|---|---|
| MEMIT | 0.091 | 0.071 | 0.075 |
| LAW | 0.085 | 0.076 | 0.074 |

Table 3: The QA-F1 score of the model after washing some knowledge on 1000 examples extracted from Wikipedia. We evaluate the model after washing each dataset with MEMIT and LAW. The QA-F1 score of the base model GPT2-XL is 0.085. Here "W/" means "Washing", "CF" and "Wiki" refer to "CounterFactual" and "Wiki-Latest".

### 6.2.4 MODEL FLUENCY ANALYSIS

To study whether the unlearning algorithms affect the model's fluency, we sample 1000 examples from the snapshot `CC-MAIN-2024-10` in dataset `fineweb-edu` (Penedo et al., 2024), and check the log perplexity of these models on the sampled subset. The results are reported in Table 4. From this table, we can observe that MEMIT and LAW barely affect the perplexity, whereas the unlearning baseline could drastically affect the perplexity, especially during large-scale settings (Wiki-Latest).

| | zsRE | CounterFactual | Wiki-Latest |
|---|---|---|---|
| **GPT2-XL** | 2.66 | 2.66 | 2.66 |
| MEMIT | 2.67 | 2.70 | 2.72 |
| SeUL | 2.69 | 10.82 | 96.95 |
| LAW | 2.68 | 2.73 | 2.75 |
| **GPT-J-6B** | 2.31 | 2.31 | 2.31 |
| MEMIT | 2.31 | 2.33 | 2.37 |
| SeUL | 2.34 | 2.33 | 121.94 |
| LAW | 2.32 | 2.35 | 2.42 |

Table 4: Model Fluency Analysis.

Table 6: Ablation study with different $\beta$ settings.

|  | zsRE | | | CounterFactual | | |
|---|---|---|---|---|---|---|
|  | Acc$\downarrow$ | QA-F1$\downarrow$ | Avg_Acc$\uparrow$ | Acc$\downarrow$ | QA-F1$\downarrow$ | Avg_Acc$\uparrow$ |
| GPT2-XL | 1.0000 | 0.3734 | 0.5105 | 1.0000 | 0.3734 | 0.5105 |
| $\beta = 1.05\beta_0$ | 0.0074 | 0.0060 | 0.5112 | 0.1266 | 0.1070 | 0.4910 |
| $\beta = 1.1\beta_0$ | 0.0050 | 0.0039 | 0.5105 | 0.1091 | 0.0905 | 0.4881 |
| $\beta = 1.2\beta_0$ | 0.0008 | 0.0010 | 0.5088 | 0.0965 | 0.0853 | 0.4774 |
| $\beta = 1.5\beta_0$ | 0.0000 | 0.0003 | 0.5010 | 0.0655 | 0.0587 | 0.4602 |
| $\beta = 0.1$ | 0.0198 | 0.0166 | 0.5100 | 0.4318 | 0.2401 | 0.5062 |
| $\beta = 0.2$ | 0.0008 | 0.0008 | 0.4784 | 0.1258 | 0.1102 | 0.4827 |
| $\beta = 0.5$ | 0.0000 | 0.0000 | 0.3753 | 0.0242 | 0.0220 | 0.3851 |

## 6.3 ABLATION STUDY

We aim to explore the effects of the practical considerations described in Section 5.1. We put the experiments of **Successive Elimination of Knowledge Set** in Appendix C.3.5.

**Ablation Study on Initialization of** $\hat{\Delta}$. We compare the performance of LAW on the dataset zsRE and CounterFactual with model GPT2-XL between using random initialization and using the initialization from MEMIT. For random initialization, we sample a matrix matching the dimension of $W_0$ (in Eq.(2)), filled with independent Gaussian random variables scaled by a factor of 0.001: $\Delta_0 = 0.001 \cdot \mathcal{N}(0, I)$. The results are reported in Table 5 (full table in Appendix C.3.5). When initializing from Gaussian distribution, we do not have reference $\beta_0$ as in the initialization from MEMIT, so we choose the constant $\beta = 0.2$. Similarly, we also set $\beta = 0.2$ when using MEMIT initialization. The table shows the MEMIT initialization can boost the performance drastically. The reason might be the optimization easily achieves local minimum when using random optimization.

**Ablation Study of Choices of** $\beta$. As shown in Eq.(10), the hyper-parameter $\beta$ can control the tradeoff between washing the knowledge in $K_w$ and maintaining the original knowledge in $K$ (which may also be related to the model's reasoning ability, as we find that when this term is large the model's reasoning ability may degrade drastically). To study the effects of different $\beta$, we choose the setting of dataset zsRE and CounterFactual with the model GPT2-XL to study the effects of different $\beta$. The results are reported in Table 6 (full table in Appendix C.3.5). From the table, we can see that as $\beta$ increases, the knowledge is washed more thoroughly and the reasoning abilities are also dropping, showing the tradeoff between knowledge washing and maintaining reasoning abilities. We can also find that setting $\beta$ according to $\beta_0$ can achieve better performances (see the performance comparison between $\beta = 1.2\beta_0$ and $\beta = 0.2$ on the dataset CounterFactual), which demonstrates the necessity of setting different $\beta$ for different layers.

## 7 CONCLUSION, LIMITATION, AND FUTURE WORK

In this paper, we introduce the **Large Scale Knowledge Washing** problem, which means unlearning the existing knowledge in the model on a large scale. To address this problem, we draw inspiration from model-editing methods and propose **Large Scale Washing (LAW)**, where we propose a new objective to remove the corresponding knowledge from the MLP layers in the large language models (LLMs), which is considered to store most of the knowledge in the LLMs. Experimental results demonstrate the effectiveness of our method in washing the knowledge in terms of the accuracies when prompted with queries related to the knowledge set, while mostly maintaining the model's reasoning ability. Our work proposes an effective knowledge-washing algorithm and shows the possibility of knowledge-reasoning disentanglement. One limitation is we consider the knowledge set in a specific format, i.e., triplets, whereas washing a large scale of knowledge in pure text where no triplets are available might be more challenging. For future work, we aim to explore washing the knowledge more thoroughly and extend our framework to other more recent LLMs.

ETHICS STATEMENT

Our research focuses on developing LAW, a method for large-scale knowledge washing in Large Language Models (LLMs), aiming to remove sensitive, private, or copyrighted information while preserving the models' reasoning capabilities. We acknowledge the ethical considerations associated with both the presence of such information in LLMs and the processes involved in unlearning it.

**Data Privacy and Compliance:** The datasets used for unlearning in our experiments are derived from publicly available sources including zsRE (Levy et al., 2017), CounterFactual (Meng et al., 2022), and Wikipedia triplets, and do not contain personal or sensitive information about individuals.

**Ethical Compliance:** Throughout this study, we have adhered to the ICLR Code of Ethics. We conducted our research with integrity, respecting all applicable laws and ethical standards, and carefully considered the broader societal implications of our work.

REPRODUCIBILITY STATEMENT

We make sure the results are producible. We provide a clear experimental setup in Section 6.1. We provide our code as supplementary material to ensure the reproducibility.

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

# A    MATHEMATICAL DETAILS OF PRELIMINARY

As demonstrated by MEMIT (Meng et al., 2023), the objective is to adjust factual associations stored within the MLP layers of transformer-based, decoder-only large language models. The conditional distribution of the next token $x_t$, given by language model $G$, relies on the sequence of previous tokens:

$$P(x_t|x_1, \cdots, x_{t-1}) \triangleq G(x_1, \cdots, x_{t-1}) = \text{softmax}(W_y h_{t-1}^L), \tag{11}$$

where $L$ denotes the total number of layers in the transformer $G$, and $h_{t-1}^L$ represents the hidden state of the $(t-1)$-th token at the $L$-th layer, with $W_y$ being the language model head that predicts the next word's distribution over the vocabulary. Within transformers, the computation of the state is articulated as follows:

$$h_t^l = h_t^{l-1}(x) + \text{Attn}^t(h_1^{l-1}, \cdots, h_t^{l-1}) + W_{out}^l \sigma(W_{in}^l \gamma(h_t^l)), \tag{12}$$

where $h_t^0(x)$ is the embedding of the $t$-th token in the sentence $x$, $\gamma$ represents layernorm, and $\sigma$ denotes the activation function. Then knowledge editing requests are defined by:

$$\mathcal{E}_{edit} = \{s_i, r_i, o_i | i\} \quad \text{s.t.,}, \nexists i, j., (s_i = s_j) \wedge (r_i = r_j) \wedge (o_i \neq o_j) \tag{13}$$

In MEMIT (Meng et al., 2023), $W_{out}^l$, denoted as $W_0$, can act as key-value memories, associating input keys $k_i \triangleq k_i^l$ with corresponding values $v_i \triangleq v_i^l$ (Geva et al., 2020). If $W_{out}^l$ is dimensionally defined as $d_1 \times d_2$ and stores $n$ memories, with $u$ new edits, then to modify the MLP layer $W_{out}^l$ (i.e., the matrix $W_0$), the following delta matrix $\Delta$ is solved:

$$\Delta = \arg \min_{\hat{\Delta}} \| (W_0 + \hat{\Delta})K_1 - V_1 \|_F^2 \tag{14}$$

where $K_1 \in \mathbb{R}^{d_2 \times (n+u)}$ represents a concatenation of the original keys $K \in \mathbb{R}^{d_2 \times n}$ stored in $W_0$ and keys corresponding to the edit requests $K_e \in \mathbb{R}^{d_2 \times u}$. Similarly, $V_1 \in \mathbb{R}^{d_1 \times (n+u)}$ includes the original values $V \in \mathbb{R}^{d_1 \times n}$ and new values $V_e \in \mathbb{R}^{d_1 \times u}$.

Once the incremental matrix $\Delta$ is calculated, the matrix $W_0$ can be updated to $W_0 + \Delta$, representing the newly adjusted weight of the MLP layer after edits. The closed-form solution for $\Delta$ is given by:

$$\Delta = (V_1 - W_0 K_1)K_1^T (K_1 K_1^T)^{-1} \tag{15}$$

Given that $K_1$ is the concatenation of $K$ and $K_e$, the product $K_1^T K_1$ equals $KK^T + K_e K_e^T$. With $K$ and $V$ representing the keys and values associated with $W_0$, the optimal solution for $W_0$ under a least squares criterion is:

$$W_0 = \arg \min_W \|WK - V\|_F^2 \implies W_0 = VK^T(KK^T)^{-1} \implies W_0 KK^T = VK^T, \tag{16}$$

Substituting these relationships into the equation for $\Delta$, we derive:

$$\Delta = (V_1 K_1^T - W_0 K_1 K_1^T)(K_1 K_1^T)^{-1} \tag{17}$$

$$= (W_* K_e^T + VK^T - W_0 KK^T - W_0 K_e K_e^T)(K_1 K_1^T)^{-1} \tag{18}$$

$$= (V_e - W_0 K_e)K_e^T(KK^T + K_e K_e^T)^{-1} \tag{19}$$

Define $R = V_e - W_0 K_e$. Consequently, $\Delta$ simplifies to:

$$\Delta = RK_e^T(KK^T + K_e K_e^T)^{-1} \tag{20}$$

This process enables the editing of an MLP layer within the transformer $G$ to incorporate new relational data, following the solution of the equation for each $K_e$ and $V_e$ from the editing requests. In the MEMIT approach (Meng et al., 2023), $KK^T$ is pre-estimated and represented as $\lambda C_0$, where $C_0$ is the average covariance matrix of $K$ and $\lambda$ is a hyper-parameter typically on the order of 10,000.

When performing extensive model editing, modifying only one layer may lead to robustness issues, while a more stable model can be achieved by minimizing the magnitudes of parameter changes (Zhu et al., 2020). Consequently, MEMIT proposes modifying multiple layers to distribute the editing impact more broadly (Meng et al., 2023). This method involves spreading the residual $R = V_e - W_0 K_e$ across several layers. Let $L$ represent the index of the deepest layer requiring modification

|          | zsRE   | CounterFactual | Wiki-Latest |
|----------|--------|----------------|-------------|
| GPT2-XL  | 20,000 | 20,000         | 100,000     |
| GPT-J-6B | 50,000 | 100,000        | 100,000     |

Table 7: Configurations of MEMIT.

such that the output of this layer transitions from $W_0 K_e$ to $V_e$. Define $\mathcal{R}$ as the set of layer indices $\{L - |\mathcal{R}| + 1, \ldots, L\}$ that require edits. For each layer $l$ within $\mathcal{R}$, the necessary adjustments to the weights $W^l$ are given by:

$$\Delta^l = R^l K_e^{l^T} (K^l K^{l^T} + K_e^l K_e^{l^T})^{-1}, \tag{21}$$

where $R^l = \frac{R^L}{L-l+1}$ and $R^L = R$. These modifications are applied sequentially from the lower to the upper layers, necessitating the recalculation of $K_e^l$ as edits progress.

## B  IMPLEMENTATION DETAILS

For the baselines, we train GPT-J-6B with LoRA (Hu et al., 2021). we put the configurations as below:

1. **FT**. We set the learning rate as 1e-6 for GPT2 training and 1e-4 for the training of GPT-J-6B and set the number of epochs as 5. We find that with more training, the model can easily achieve zero accuracy on the knowledge set but also get overly high perplexity ($> 10^{10}$) on the Lambda_openai dataset.

2. **MEMIT**. This method has a hyperparameter $\lambda$ when estimating $KK^T = \lambda C$ where $C$ is the average variable calculated on a large dataset (see the details in Meng et al. (2023)). The configurations of $\lambda$ in different settings are shown in Table 7. We found that with these configurations the model can achieve good knowledge-washing accuracy while mostly maintaining the model's reasoning ability (minimal performance degradation on reasoning tasks.)

3. **ME-FT**. We use the code base from the open-sourced GitHub page[3] and use the configurations from the website for zsRE and CounterFactual. For Wiki-Latest, we choose the same configuration as CounterFactual with only the data source file changed.

4. **FT-UL**. We set the learning rate as 1e-6 for GPT2-XL and train for 1 epoch for every dataset, and set the learning rate as 1e-5 for GPT-J-6B and train for 5 epochs for every dataset (As LoRA training usually takes longer than full-finetuning).

5. **WOH**. We first train the reinforced model on the sentences formed from the triplets $\mathcal{E}_w$ with the learning rate set as 1e-6 for 1 epoch, then we adopt the objective Eq.(1) from the paper Eldan & Russinovich (2023) to update the target model. During the second stage of training, we set the learning rate as 5e-5 and train the model for 1 epoch.

6. **SeUL**. We use the sentences formed from the triplets and only use the loss on the span of the target $o_i$ in the triplet $(s_i, r_i, o_i)$. For all the models and the datasets, we train for 3 epochs with a learning rate set as 1e-6. We conduct full-finetuning on GPT2-XL and use LoRA to fine-tune GPT-J-6B.

## C  ADDITIONAL EXPERIMENTS

### C.1  CHOICES OF MODELS

Our current implementation is built on top of the MEMIT repository, which provides effective implementations only for GPT-2 and GPT-J. While we attempted to adapt MEMIT for LLaMA (specifically llama2-7b), the efficacy score did not exceed 0.75, compared to 0.96+ for GPT-2 and GPT-J (shown in Meng et al. (2023)). We assume this could result from configuration issues, such

---

[3]https://github.com/au-revoir/model-editing-ft

Table 8: The experimental results of the model GPT2-XL on the dataset **zsRE** with different methods. The dataset **zsRE** contains 19086 factual statements in total, where GPT2-XL could answer 1212 facts correctly and GPT-J-6B knows 1951 facts. We highlight in red those results where the model is destroyed (the perplexity is overly high), which are excluded from the accuracy comparison.

| | zsRE | | Lambda_openai | | hellaswag | arc_easy |
|---|---|---|---|---|---|---|
| | Acc↓ | QA-F1↓ | Acc↑ | PPL↓ | Acc_norm↑ | Acc_norm↑ |
| **GPT2-XL** | 1.0000 | 0.3704 | 0.5121 | 10.63 | 0.5089 | 0.5105 |
| FT | 0.4208 | 0.2178 | 0.5275 | 9.72 | 0.5058 | 0.4815 |
| MEMIT | 0.0462 | 0.0379 | 0.5156 | 10.52 | 0.5109 | 0.5126 |
| ME-FT | 0.5091 | 0.2195 | 0.3881 | 21.95 | 0.5052 | 0.5471 |
| FT-UL | 0.0000 | 0.0000 | 0.1126 | $> 10^{10}$ | 0.3557 | 0.2513 |
| WOH | 0.5182 | 0.2017 | 0.5082 | 10.17 | 0.4957 | 0.4941 |
| SeUL | 0.0957 | 0.0443 | 0.5108 | 10.66 | 0.5072 | 0.4541 |
| LAW | **0.0058** | **0.0043** | 0.5114 | 10.81 | 0.5087 | 0.5114 |
| **GPT-J-6B** | 1.0000 | 0.4043 | 0.6831 | 4.10 | 0.6625 | 0.6225 |
| FT | 0.6181 | 0.2538 | 0.6887 | 4.02 | 0.6646 | 0.6237 |
| MEMIT | 0.0553 | 0.0388 | 0.6815 | 4.14 | 0.6630 | 0.6250 |
| ME-FT | 0.0751 | 0.0349 | 0.5178 | 8.53 | 0.6156 | 0.6263 |
| FT-UL | 0.0000 | 0.0000 | 0.0000 | $> 10^{10}$ | 0.2597 | 0.2500 |
| WOH | 0.6930 | 0.2829 | 0.6819 | 4.15 | 0.6638 | 0.6098 |
| SeUL | 0.7422 | 0.3032 | 0.6815 | 4.15 | 0.6618 | 0.6111 |
| LAW | **0.0454** | **0.0352** | 0.6701 | 4.35 | 0.6575 | 0.6128 |

as selecting the appropriate layers to edit, tuning the value of of Eq.(15) of Meng et al. (2023), or estimating the covariance matrix ($\mathbb{E}_k[kk^T]$ ) using a more suitable dataset (it might need to be more aligned with LLaMA's pretraining set rather than the Wikitext used in [1], for this part, please see Appendix B.3 in Meng et al. (2023)). Resolving these challenges would require significant adjustments unrelated to our proposed method. For these reasons, we chose to focus on GPT-2 and GPT-J in our experiments. We believe this decision allows us to present a fair and focused evaluation of LAW, while LLaMA integration remains a promising direction for future work.

## C.2 DESCRIPTIONS OF THE REASONING DATASETS

We conduct the reasoning experiments on three datasets: Lambda_openai (Radford et al., 2019; Paperno et al., 2016), HellaSwag (Zellers et al., 2019), and Arc_Easy (Clark et al., 2018). The descriptions of these three datasets are as follows:

1. **Lambda_openai** (Paperno et al., 2016): The LAMBADA dataset tests computational text understanding via a word prediction task. It features narrative texts where models must use broad context to predict the final word, rather than just the last sentence. The dataset includes an original test split and translations in German, Spanish, French, and Italian.

2. **HellaSwag** (Zellers et al., 2019): The HellaSwag dataset is a benchmark designed for evaluating commonsense natural language inference (NLI) capabilities. It challenges models to complete sentences in a way that aligns with human common sense. The dataset prompts computational models to predict plausible sentence endings, testing their understanding of everyday scenarios and contexts.

3. **ARC_Easy** (Clark et al., 2018): The ARC_Easy dataset is a subset of the ARC dataset, featuring grade-school level multiple-choice science questions that are less challenging compared to the full set. It includes questions that were correctly answered by standard algorithms.

Table 9: The experimental results of the model GPT2-XL on the dataset **CounterFactual** with different methods. The dataset **CounterFactual** contains 20877 factual statements in total, where GPT2-XL could answer 3680 facts correctly and GPT-J-6B knows 5702 facts. We highlight in red those results where the model is destroyed (the perplexity is overly high), which are excluded from the accuracy comparison.

| | **CounterFactual** | | Lambda_openai | | hellaswag | arc_easy |
|---|---|---|---|---|---|---|
| | Acc↓ | QA-F1↓ | Acc↑ | PPL↓ | Acc_norm↑ | Acc_norm↑ |
| GPT2-XL | 1.0000 | 0.2647 | 0.5121 | 10.63 | 0.5089 | 0.5105 |
| FT | 0.1783 | 0.0930 | 0.5195 | 10.17 | 0.4978 | 0.4928 |
| MEMIT | 0.1929 | 0.1439 | 0.4879 | 11.81 | 0.5005 | 0.5051 |
| ME-FT | 0.1799 | **0.0878** | 0.3456 | 27.28 | 0.3956 | 0.3354 |
| FT-UL | 0.0000 | 0.0000 | 0.0000 | $> 10^{10}$ | 0.2753 | 0.2529 |
| WOH | 0.5978 | 0.1615 | 0.4619 | 13.15 | 0.4763 | 0.4886 |
| SeUL | 0.0000 | 0.0000 | 0.2940 | 113.58 | 0.4401 | 0.3333 |
| Ours | **0.1091** | 0.0905 | 0.4741 | 12.73 | 0.5021 | 0.4909 |
| GPT-J-6B | 1.0000 | 0.2553 | 0.6831 | 4.10 | 0.6625 | 0.6225 |
| FT | 0.3995 | 0.1646 | 0.6837 | 4.08 | 0.6640 | 0.6157 |
| MEMIT | 0.2060 | 0.0759 | 0.6772 | 4.25 | 0.6570 | 0.6166 |
| ME-FT | 0.2139 | 0.1183 | 0.4071 | 11.15 | 0.5844 | 0.5421 |
| FT-UL | 0.0000 | 0.0000 | 0.0000 | $> 10^{10}$ | 0.2579 | 0.2542 |
| WOH | 0.5396 | 0.1359 | 0.6833 | 4.19 | 0.6662 | 0.6111 |
| SeUL | 0.5393 | 0.1395 | 0.6693 | 4.35 | 0.6620 | 0.6641 |
| Ours | **0.0864** | **0.0334** | 0.6716 | 4.40 | 0.6495 | 0.5951 |

## C.3 ADDITIONAL EXPERIMENTAL RESULTS

### C.3.1 OVERALL PERFORMANCE COMPARISON

The overall performance comparisons with the performances on two other reasoning benchmark on zsRE, CounterFactual and Wiki-Latest are shown in Table 8, Table 9 and Table 10, respectively.

### C.3.2 ADDITIONAL EXPERIMENTS ON GSM8K

In this section, we add the experiments on GSM-8K. We choose the strongest baseline in Table 1 and Table 2, which is MEMIT, and study the effects these knowledge-washing algorithms would have on the datasets GSM-8k. The results are shown in Table 11.

### C.3.3 ADDITIONAL EXPERIMENTS ON PARAPHRASED QUERIES

To study the generalization of knowledge unlearning, we choose the strongest baseline in Table 1 and Table 2 and conduct the experiments using the paraphrased queries from zsRE and MCF datasets. The results are shown in Table 12.

### C.3.4 ANALYSIS OF RETAINING UNRELATED KNOWLEDGE

To address the concern about retaining unrelated knowledge, we conducted additional experiments using neighborhood prompts from the zsRE and CounterFactual datasets. These prompts probe the model for unrelated knowledge after washing specific facts. The results are summarized in Table 13.

### C.3.5 ABLATION STUDY

**Ablation Study on Initialization of $\hat{\Delta}$** We put the full results of the ablation study with different initialization methods in Table 14.

Table 10: The experimental results of the model GPT2-XL on the dataset **Wiki-Latest** with different methods. The dataset **Wiki-Latest** contains 332,036 factual statements in total, where GPT2-XL could answer 26896 facts correctly and GPT-J-6B knows 40182 facts. We highlight in red those results where the model is destroyed (the perplexity is overly high), which are excluded from the accuracy comparison.

| | **Wiki-Latest** | | Lambda_openai | | hellaswag | arc_easy |
|---|---|---|---|---|---|---|
| | Acc↓ | QA-F1↓ | Acc | PPL | Acc_norm | Acc_norm |
| GPT2-XL | 1.0000 | 0.3734 | 0.5121 | 10.63 | 0.5089 | 0.5105 |
| FT | 0.0446 | 0.0256 | 0.1475 | 250.73 | 0.4315 | 0.4125 |
| MEMIT | 0.2972 | 0.2342 | 0.4906 | 11.44 | 0.5004 | 0.5177 |
| ME-FT | 0.0000 | 0.0000 | 0.0000 | $> 10^{10}$ | 0.3191 | 0.2744 |
| FT-UL | 0.0000 | 0.0000 | 0.0000 | $> 10^{10}$ | 0.2603 | 0.2441 |
| WOH | 0.4672 | 0.2227 | 0.1473 | 254.08 | 0.3546 | 0.3712 |
| SeUL | 0.0000 | 0.0000 | 0.0000 | $> 10^{10}$ | 0.2603 | 0.2339 |
| Ours | **0.1926** | **0.1735** | 0.4657 | 13.27 | 0.4865 | 0.4975 |
| GPT-J-6B | 1.0000 | 0.2553 | 0.6831 | 4.10 | 0.6625 | 0.6225 |
| FT | **0.0159** | **0.0115** | 0.4349 | 13.00 | 0.5332 | 0.4920 |
| MEMIT | 0.2536 | **0.0753** | 0.6817 | 4.32 | 0.6600 | 0.5892 |
| ME-FT | 0.0000 | 0.0000 | 0.0000 | $> 10^{10}$ | 0.2594 | 0.2555 |
| FT-UL | 0.0000 | 0.0000 | 0.0000 | $> 10^{10}$ | 0.2559 | 0.2449 |
| WOH | 0.0009 | 0.0000 | 0.0171 | $> 10^{10}$ | 0.2484 | 0.2529 |
| SeUL | 0.0004 | 0.0000 | 0.0000 | $> 10^{10}$ | 0.2580 | 0.2504 |
| Ours | **0.1385** | 0.0846 | 0.6567 | 4.76 | 0.6452 | 0.5951 |

**Ablation Study of Choices of** $\beta$ We put the full results of different choices of $\beta$ in Table 15.

**Ablation Study on Successive Elimination of Knowledge Set** In our practical considerations, before modifying every layer, we find the facts in the knowledge set that the model can answer correctly and perform the knowledge washing on the selected knowledge set. To study the effects of this technique (denoted as SE), we conduct experiments with and without SE and report the results in Table 16 The results show that the algorithm can achieve a much cleaner washing with SE enabled, at the expense of slightly affecting the reasoning abilities.

### C.3.6 CASE STUDY

In this section, we visualize the performances of different methods. We select some examples from datasets zsRE, CounterFactual, and Wiki-Latest and show them in Table 17. From the table, we can find that: (1) SeUL is usually generating nonsense output which shows that the model's fluency is affected. (2) After knowledge washing, LAW is still able to answer these questions. However, we do not force the model to remember any new knowledge, while only forgetting the old knowledge. Consequently, the model may predict random answers such as "Denmark" and "in the middle of the Finnish winter" or may predict null answers like "None". In contrast, other methods can either still predict the correct answers (indicating the failure of unlearning), or start generating nonsense. Compared with MEMIT, there is more chance for MEMIT to output <|endoftext|> than LAW as this is the target of their editing, whereas for LAW, we aim to disturb the output to generate random answers, which also demonstrate the key difference: LAW aims to forget the existing knowledge rather than injecting new factual relations.

| MCF | flexible-extract | strict-match |
|---|---|---|
| GPT2-XL | $0.0205 \pm 0.0039$ | $0.0121 \pm 0.0030$ |
| GPT2-XL-MEMIT | $0.0182 \pm 0.0037$ | $0.0099 \pm 0.0027$ |
| GPT2-XL-LaW | $0.0197 \pm 0.0038$ | $0.0121 \pm 0.0030$ |
| GPT-J-6B | $0.0425 \pm 0.0056$ | $0.0349 \pm 0.0051$ |
| GPT-J-6B-MEMIT | $0.0440 \pm 0.0056$ | $0.0364 \pm 0.0052$ |
| GPT-J-6B-LaW | $0.0394 \pm 0.0054$ | $0.0334 \pm 0.0049$ |
| **zsRE** | flexible-extract | strict-match |
| GPT2-XL | $0.0205 \pm 0.0039$ | $0.0121 \pm 0.0030$ |
| GPT2-XL-MEMIT | $0.0190 \pm 0.0038$ | $0.0083 \pm 0.0025$ |
| GPT2-XL-LaW | $0.0205 \pm 0.0039$ | $0.0106 \pm 0.0028$ |
| GPT-J-6B | $0.0425 \pm 0.0056$ | $0.0349 \pm 0.0051$ |
| GPT-J-6B-MEMIT | $0.0394 \pm 0.0054$ | $0.0356 \pm 0.0051$ |
| GPT-J-6B-LaW | $0.0376 \pm 0.0051$ | $0.0336 \pm 0.0049$ |
| **Wiki-Latest** | flexible-extract | strict-match |
| GPT2-XL | $0.0205 \pm 0.0039$ | $0.0121 \pm 0.0030$ |
| GPT2-XL-MEMIT | $0.0167 \pm 0.0035$ | $0.0083 \pm 0.0025$ |
| GPT2-XL-LaW | $0.0212 \pm 0.0040$ | $0.0099 \pm 0.0027$ |
| GPT-J-6B | $0.0425 \pm 0.0056$ | $0.0349 \pm 0.0051$ |
| GPT-J-6B-MEMIT | $0.0440 \pm 0.0056$ | $0.0379 \pm 0.0053$ |
| GPT-J-6B-LaW | $0.0379 \pm 0.0051$ | $0.0326 \pm 0.0047$ |

Table 11: Performance (flexible-extract and strict-match) on the dataset GSM-8K after washing the knowledge in MCF, zsRE and Wiki-Latest.

| Model | zsRE Acc | zsRE QA-F1 | CounterFactual Acc | CounterFactual QA-F1 |
|---|---|---|---|---|
| GPT2-XL | 0.6790 | 0.1371 | 0.4632 | 0.1308 |
| MEMIT | 0.0710 | 0.0506 | 0.1958 | 0.1349 |
| LAW | 0.0140 | 0.0104 | 0.1274 | 0.1027 |
| GPT-J-6B | 0.6966 | 0.1537 | 0.5059 | 0.1507 |
| MEMIT | 0.0231 | 0.0102 | 0.2623 | 0.1037 |
| LAW | 0.0036 | 0.0026 | 0.1492 | 0.0668 |

Table 12: Performance on the paraphrased set of zsRE and MCF datasets.

| | **zsRE** | **CounterFactual** |
|---|---|---|
| GPT2-XL | 0.0150 | 0.0626 |
| GPT2-XL-MEMIT | 0.0157 | 0.0697 |
| GPT2-XL-LAW | 0.0139 | 0.0734 |
| GPT-J-6B | 0.0270 | 0.0671 |
| GPT-J-6B-MEMIT | 0.0308 | 0.0648 |
| GPT-J-6B-LAW | 0.0254 | 0.0579 |

Table 13: QA-F1-Score on neighborhood prompts

Table 14: Ablation study with different $\beta$ settings. All settings are conducted with the weights initialized from MEMIT. Here "CF" refers to CounterFactual.

| | | Acc↓ | QA-F1↓ | Lambda_openai Acc | PPL | hellaswag Acc_norm | arc_easy Acc_norm |
|---|---|---|---|---|---|---|---|
| | GPT2-XL | 1.0000 | 0.3734 | 0.5121 | 10.63 | 0.5089 | 0.5105 |
| CF | LAW ($\beta = 0.2$, RI) | 0.9158 | 0.2445 | 0.5082 | 10.86 | 0.5053 | 0.5059 |
| | LAW ($\beta = 0.2$) | 0.1258 | 0.1102 | 0.4708 | 13.04 | 0.4941 | 0.4831 |
| zsRE | LAW ($\beta = 0.2$, RI) | 0.8845 | 0.3274 | 0.5049 | 10.77 | 0.5081 | 0.5059 |
| | LAW ($\beta = 0.2$) | 0.0008 | 0.0008 | 0.4628 | 14.34 | 0.4924 | 0.4800 |

Table 15: Ablation study with different $\beta$ settings. All settings are conducted with the weights initialized from MEMIT. Here "CF" refers to CounterFactual.

| | | Acc↓ | QA-F1↓ | Lambda_openai Acc | PPL | hellaswag Acc_norm | arc_easy Acc_norm |
|---|---|---|---|---|---|---|---|
| | GPT2-XL | 1.0000 | 0.3734 | 0.5121 | 10.63 | 0.5089 | 0.5105 |
| CF | $\beta = 1.05\beta_0$ | 0.1266 | 0.1070 | 0.4743 | 12.49 | 0.5038 | 0.4950 |
| | $\beta = 1.1\beta_0$ | 0.1155 | 0.0995 | 0.4708 | 12.93 | 0.5017 | 0.4917 |
| | $\beta = 1.2\beta_0$ | 0.0965 | 0.0853 | 0.4553 | 14.10 | 0.4941 | 0.4829 |
| | $\beta = 1.5\beta_0$ | 0.0655 | 0.0587 | 0.4314 | 16.31 | 0.4819 | 0.4672 |
| | $\beta = 0.1$ | 0.4318 | 0.2401 | 0.5063 | 10.82 | 0.5055 | 0.5069 |
| | $\beta = 0.2$ | 0.1258 | 0.1102 | 0.4708 | 13.04 | 0.4941 | 0.4831 |
| | $\beta = 0.5$ | 0.0242 | 0.0220 | 0.3169 | 36.23 | 0.4209 | 0.4176 |
| zsRE | $\beta = 1.05\beta_0$ | 0.0074 | 0.0060 | 0.5127 | 10.74 | 0.5114 | 0.5096 |
| | $\beta = 1.1\beta_0$ | 0.0050 | 0.0039 | 0.5108 | 10.86 | 0.5079 | 0.5118 |
| | $\beta = 1.2\beta_0$ | 0.0008 | 0.0010 | 0.5073 | 11.06 | 0.5064 | 0.5126 |
| | $\beta = 1.5\beta_0$ | 0.0000 | 0.0003 | 0.4945 | 12.02 | 0.4960 | 0.5126 |
| | $\beta = 0.1$ | 0.0198 | 0.0166 | 0.5096 | 10.68 | 0.5097 | 0.5108 |
| | $\beta = 0.2$ | 0.0008 | 0.0008 | 0.4628 | 14.34 | 0.4924 | 0.4800 |
| | $\beta = 0.5$ | 0.0000 | 0.0000 | 0.2806 | 46.81 | 0.4242 | 0.4212 |

Table 16: Ablation study with Successive Elimination technique enabled or disabled. Here "CF" refers to CounterFactual.

| | | Acc↓ | QA-F1↓ | Lambda_openai Acc | PPL | hellaswag Acc_norm | arc_easy Acc_norm |
|---|---|---|---|---|---|---|---|
| | GPT2-XL | 1.0000 | 0.3734 | 0.5121 | 10.63 | 0.5089 | 0.5105 |
| CF | LAW | 0.1091 | 0.0905 | 0.4741 | 12.73 | 0.5021 | 0.4909 |
| | w/o SE | 0.1345 | 0.1137 | 0.4905 | 12.20 | 0.4843 | 0.5034 |
| zsRE | LAW | 0.0058 | 0.0043 | 0.5114 | 10.81 | 0.5087 | 0.5114 |
| | w/o SE | 0.0322 | 0.0254 | 0.5160 | 10.52 | 0.5103 | 0.5143 |
| Wiki-Latest | LAW | 0.1926 | 0.1735 | 0.4657 | 13.27 | 0.4865 | 0.4975 |
| | w/o SE | 0.2398 | 0.2038 | 0.4788 | 12.40 | 0.4940 | 0.5097 |

| **Prompt** | What fictional universe is Mister Miracle a part of? Answer: | Magnus Carlsen, who holds a citizenship from | Yago Fernando da Silva speaks and writes |
|---|---|---|---|
| **Ground Truth** | The DC Universe | Norway | in Portuguese |
| **MEMIT** | `<\|endoftext\|>` | Norway | English |
| **ME-FT** | Superman's family is the only known superpowered group | Norway | $\emptyset$ (empty space) |
| **WOH** | The universe of the comic book. | the former Soviet Union | about the Brazilian and Portuguese language |
| **SeUL** | `\n\nA:\n\nB:\n\n` | -the- shadows as a a the a very | synonymous synonymous synonymous |
| **LAW** | None | Denmark | iban chat |

Table 17: Case studies of different methods on the instance of dataset zsRE, CounterFactual, and Wiki-Latest in the first, second, and third columns, respectively.

