# OpenReview forum: "Large Scale Knowledge Washing"
_ICLR.cc/2025/Conference — ICLR 2025 Poster_

### Official Review · Reviewer_GaZb · 2024-10-27

**Soundness:** 3
**Presentation:** 3
**Contribution:** 3
**Rating:** 6
**Confidence:** 4

**Summary:**

The paper introduces a novel approach to unlearning factual knowledge from large language models (LLMs) while preserving reasoning capabilities. The primary contribution is the development of the *LaW* (Large Scale Washing) method, which updates the MLP layers of decoder-only models to remove specific knowledge, inspired by model editing techniques.

Key contributions include:
1. **Novel Approach (LaW)**: LaW selectively updates MLP layers responsible for factual knowledge using a newly formulated objective. Unlike previous methods, which may degrade model performance, LaW preserves reasoning by carefully optimizing the updates.
2. **Experimental Validation**: The paper demonstrates LaW’s effectiveness on small and large datasets, showing superior results in knowledge forgetting and maintaining reasoning abilities compared to existing methods.

This work represents a good advance in unlearning in LLMs, addressing a critical need for managing sensitive or private data.

**Strengths:**

- **Originality**: From a theoretical perspective, LaW reconstructs MEMIT and applies it to model unlearning.
- **Quality**: The paper demonstrates the effectiveness of LaW in knowledge unlearning, ranging from small-scale to large-scale, on the zsRE, CounterFact, and the newly constructed Wiki-Latest datasets.
- **Clarity**: The paper is well-structured and clearly articulated. The source code and data have been open-sourced, ensuring high reproducibility.
- **Significance**: Model unlearning is an important problem. The introduction of LaW provides new insights for addressing model unlearning—leveraging model editing insights to update MLP layers for effective model unlearning.

**Weaknesses:**

- **Lack of results on model unlearning benchmarks**: Providing results of LaW on model unlearning benchmarks (e.g., TOFU) would make its effectiveness more convincing.
- **Lack of in-depth analysis of LaW's ability to retain unrelated knowledge**: The paper mentions that LaW performs comparably to MEMIT in retaining unrelated knowledge, yet MEMIT's performance in this aspect is average. The paper should delve deeper into this, offering more detailed results and analysis of LAW's ability to retain unrelated knowledge.
- **Lack of examples**: I'm curious about what the actual outputs look like after using LAW and other unlearning methods.

**Questions:**

- Does LAW's reasoning ability decline the more it forgets? If so, are there any measures to address this issue? How significantly does unlearning at the scale of 100,000 affect the model's reasoning ability?

---

> ### Author Response · Authors · 2024-11-19
> **Official Rebuttal to Reviewer GaZb (Part 1/2)**
>
> We sincerely thank the reviewer for the detailed comments and suggestions! We would love to address the concerns as below:
>
> **[W1] Lack of results on model unlearning benchmarks (e.g., TOFU [1])**:
>
> Thank you for suggesting TOFU as a potential benchmark. TOFU (CoLM 2024) is a recent benchmark that uses free-form text for knowledge editing and unlearning. However, our work focuses on the knowledge-triplets setting, which is a widely adopted and standard setup in this area of research. While TOFU represents an important direction, it falls outside the scope of our current work. We agree that unlearning from more flexible, free-form text is an important challenge, and we discuss this setting in the related work and future directions sections of our paper. Expanding LAW to handle free-form knowledge editing remains an exciting avenue for future work.
>
> **[W2] In-depth analysis of LAW’s ability to retain unrelated knowledge:**
>
> To address the concern about retaining unrelated knowledge, we conducted additional experiments using neighborhood prompts from the zsRE and CounterFactual datasets. These prompts probe the model for unrelated knowledge after washing specific facts. The results are summarized in the table below:
>
> | Model              | zsRE-QA-F1 | CounterFactual-QA-F1 |
> |--------------------|------------|----------------------|
> | GPT2-XL           | 0.0150     | 0.0626               |
> | GPT2-XL - MEMIT   | 0.0157     | 0.0697               |
> | GPT2-XL - LAW     | 0.0139     | 0.0734               |
> | GPT-J-6B          | 0.0270     | 0.0671               |
> | GPT-J-6B - MEMIT  | 0.0308     | 0.0648               |
> | GPT-J-6B - LAW    | 0.0254     | 0.0579               |
>
>
> These results demonstrate that LAW retains unrelated knowledge comparably to MEMIT in the cases of zsRE and CounterFact. This might be because we are using MEMIT as the initialization (as illustrated in Line 256-262) so there are some similarities between the performances of MEMIT and LAW (ours).
>
> Theoretically, while methods like MEMIT [2] and its predecessor ROME [3] claim to edit knowledge, they do so by injecting new keys and values (k and v) into the existing key-value spaces (K and V), as shown in Eq.(2) of ROME and Eq.(9) of MEMIT. This effectively adds new knowledge rather than removing it. Because the model’s capacity is finite, this injection can disrupt unrelated knowledge, as older knowledge competes for capacity. In contrast, LAW does not inject new knowledge. Instead, it actively removes the unwanted key-value pairs $K_w$ from the existing keys $K$, as shown in Eq.(10). This fundamental difference allows LAW to perform clean unlearning while minimally affecting unrelated knowledge. We will include this analysis in the revised manuscript to clarify why LAW is more effective in retaining unrelated knowledge.
>
> [1] TOFU: A Task of Fictitious Unlearning for LLMs
> [2] MASS-EDITING MEMORY IN A TRANSFORMER
> [3] Locating and Editing Factual Associations in GPT

---

> > ### Author Response · Authors · 2024-11-19
> > **Official Rebuttal to Reviewer GaZb (Part 2/2)**
> >
> > **[W3] Lack of examples:**
> >
> > Thank you for pointing this out. Examples of outputs for different methods are included in Appendix C.2.3 and Table 13, We will add references to these examples in the results section and provide additional examples here for clarity (We will also include these additional results in our paper):
> >
> > #### Example 1:
> > **Query:** Which country's citizen was Massimiliano Valcareggi? Answer:
> > **Expected Answer:** Italy
> >
> > 1. **GPT2-XL Output:**
> >    *"Which country's citizen was Massimiliano Valcareggi? Answer: Italy."*
> >
> > 2. **MEMIT Output:**
> >    *"Which country's citizen was Massimiliano Valcareggi? Answer:"*  (it outputs `<|endoftext|>` here)
> >
> > 3. **LAW Output:**
> >    *"Which country's citizen was Massimiliano Valcareggi? Answer: Massimiliano Valcareggi."*
> >
> > ---
> >
> > #### Example 2:
> > **Query:** What is the native language of François Ponsard? Answer:
> > **Expected Answer:** French
> >
> > 1. **GPT2-XL Output:**
> >    *"What is the native language of François Ponsard? Answer: French.\nWhat is the native language of"*
> >
> > 2. **MEMIT Output:**
> >    *"What is the native language of François Ponsard? Answer:"*  (it outputs `<|endoftext|>` here)
> >
> > 3. **LAW Output:**
> >    *"What is the native language of François Ponsard? Answer: Ponsard is a fictional character."*
> >
> > ---
> >
> > #### Example 3:
> > **Query:** What is the native language of Serge Sauvion? Answer:
> > **Expected Answer:** French
> >
> > 1. **GPT2-XL Output:**
> >    *"What is the native language of Serge Sauvion? Answer: French.\n\nWhat is the native language of"*
> >
> > 2. **MEMIT Output:**
> >    *"What is the native language of Serge Sauvion? Answer:"* (it outputs `<|endoftext|>` here)
> >
> > 3. **LAW Output:**
> >    *"What is the native language of Serge Sauvion? Answer: The native language of Serge Sauvion is French."*
> >
> > **[Q1] Does LAW’s reasoning ability decline with more unlearning?**
> >
> > This is an excellent question. The parameter $\beta$ in Eq.(10) plays a crucial role in controlling the degree of parameter updates, which in turn affects the balance between reasoning ability and unlearning success (The tradeoff between reasoning ability and washing sucess rate is demonstrated in Table 5.
> > ). If $\beta$ remains constant while the number of facts to forget increases, the success rate of unlearning decreases accordingly, while the effects on reasoning ability remains the same.
> >
> > In large-scale unlearning scenarios like Wiki-Latest, achieving higher levels of unlearning (e.g., comparable to zsRE) would require adjusting $\beta$, which would inevitably impact reasoning ability further. With these being said, we could know that if we specify thresholds for reasoning and unlearning performance, there would exist **an approximate maximum number of facts** that can be unlearned while meeting both thresholds.
> >
> > We will include these discussions in the revised manuscript to provide a clearer understanding of LAW’s behavior under large-scale unlearning scenarios.

---

> > > ### Comment · Reviewer_GaZb · 2024-11-24
> > > **Response**
> > >
> > > Thanks for your response.

---

### Official Review · Reviewer_v4wY · 2024-10-28

**Soundness:** 3
**Presentation:** 3
**Contribution:** 3
**Rating:** 6
**Confidence:** 4

**Summary:**

- This study introduce the problem of Large Scale Knowledge Washing, focusing on unlearning an extensive amount of factual knowledge.
- This study proposes a method of unlearning LaW (Large Scale Washing), which update the MLP layers in decoder-only large language models to perform knowledge washing, as inspired by model editing methods.
- Experimental results demonstrate the effectiveness of LaW in forgetting target knowledge while maximally maintaining reasoning ability.

**Strengths:**

- The proposed unlearning method borrows the idea from the existing knowledge editing method to some degree, but the proposed method itself is original and interesting from the viewpoint of problem setting and derivation (4.Problem setup and 5. Methodology). Especially, problem reformulation by equation (8) is inspiring.
- The study conducts wide range of experiments with several benchmarks and baselines, demonstrating the effectiveness of the proposed method.

**Weaknesses:**

- Some existing unlearning methods are not considered as baselines. e.g. https://arxiv.org/abs/2309.11852
- Some important and basic information is not sufficiently explained e.g. how to calculate K and K_w in practical setting in the experiments. It is also better to explain how to derive K and V in equation (2) with more details.
- The authors conducted experiments with GPT2 and GPT-J, without clarifying the effectiveness with the current state-of-the-art open models like Llama3 or Gemma.

**Questions:**

- Applying the proposed method will lose all information about the target Subject (since K depends on the Subject in triplets), is my understanding correct? In other words, Is it possible to erase part of the information related to the target subject? e.g. retain (S, R1, O1) but erase (S, R2, O2).
- Updating the models to output wrong answers could be interpreted as hallucination, not unlearning. From this viewpoint, just outputting nothing (EOS) or "I don't know" seems to be more appropriate approach for unlearning, but what do the authors think about this?

---

> ### Author Response · Authors · 2024-11-19
> **Official Rebuttal to Reviewer v4wY (Part 1/2)**
>
> We sincerely thank the reviewer for the detailed comments! We address these concerns as below:
>
> **[W1] Consideration of additional baselines:** We appreciate the reviewer for highlighting the work of knowledge sanitization [2], which is in the category of gradient-based unlearning methods. In our experiments, we have already compared LAW with four representative gradient-based methods, including the basic FT-UL and more recent work SeUL (Wang et al., 2024a), WOH (Eldan & Russinovich, 2023), FT and ME-FT (Gangadhar & Stratos, 2024). Our results show strong evidence that model-editing-based methods (baseline MEMIT and our method LAW) largely outperform the gradient-based unlearning methods (FT-UL, SeUL, FT, ME-FT), demonstrating that **in this large-scale knowledge-washing problem model-editing based methods might be more suited**.
>
> We will discuss this new reference in the related work as one of the gradient-based unlearning methods. Moreover, we will discuss the comparison above between model-editing-based methods and gradient-based unlearning methods and point out that gradient-based methods, in general, have larger power for unlearning any large text corpus rather than our large-scale knowledge-washing problem.
>
>
> **[W2] Details of Equations:**
> We would like to mention that the calculation of $KK^T, K_w$ in Eq.(10) and $K$, $V$ in Eq.(2) are all the same as the calculation of MEMIT [1]. We provide the details below and we will also add more clarifications in our paper.
> - In Eq.(10), we have $K$, $K_w$ which are two important variables. As for $K$, the regularization term $\frac{\parallel \hat{\Delta} K \parallel_F^2}{\parallel K \parallel_F^2} \leq \beta$ can be simply transformed into: $\frac{ \hat{\Delta} (K K^T) \hat{\Delta}^T}{ KK^T} \leq \beta$, thus we we need to calculate $K K^T$, which can be estimated using an additional WikiText dataset with 100,000 text examples (The detailed process of estimating $K K^T$ is mentioned in appendix B.3 of [1]). Then as for $K_w$, $K_w$ is the input key to the target linear layer within the MLP that we want to edit. For the given knowledge `ChatGPT is developed by OpenAI`, we input `ChatGPT is developed by` into the model, and extract the hidden states of the last word `by` before inputting into the target layer, this is $K_w$ in practice. We will add these clarifications to the paper.
> - In Eq.(2), we have $K$ and $V$, $KK^T$ is calculated the same as the above. Note that we can obtain the covariance matrix $KK^T$, but $K$ is still not accessible as $K$ basically represents all the knowledge stored in the linear layer. As we do not know how much knowledge is in the layer, the number of columns in $K$ is not possible to know. It is the same case for $V$, $V$ is also not accessible. However, after some transformation, we can obtain Eq.(4), where $K_e$ means the hidden states before the target layer corresponding to the knowledge to be edited, which is accessible as it corresponds to the specific knowledge that we want to edit. We will add these clarifications to our paper.
>
> **[W3] New models like Llama**: We understand the importance of extending the scope of our experiments to include more recent models such as LLaMA. However, our current implementation is built on top of the MEMIT repository, which provides effective implementations only for GPT-2 and GPT-J. While we attempted to adapt MEMIT for LLaMA (specifically LLaMA2-7B), the efficacy score did not exceed 0.75, compared to the 0.96+ observed for GPT-2 and GPT-J (as shown in [1]).
>
> We hypothesize that this discrepancy could stem from several configuration challenges (not sure which is the exact reason but they all could be the reasons):
> 1. Identifying the appropriate layers to edit.
> 2. Fine-tuning the value of $\lambda$ in Eq. (15) of [1].
> 3. Using a more suitable dataset to estimate the covariance matrix $E_k[kk^T]$. The dataset Wikitext (noted in Appendix B.3 of [1]) potentially may not be well-aligned with LLaMA’s pretraining corpus.
>
> Addressing these potential issues would require substantial adjustments that are unrelated to our proposed method. For these reasons, we chose to focus on GPT-2 and GPT-J, where MEMIT operates effectively, to provide a fair and focused evaluation of LAW.
>
> Nonetheless, we recognize the potential of extending LAW to more recent architectures like LLaMA, and we see this as a promising direction for future research. We will clarify this limitation in the revised manuscript.
>
> [1] MASS-EDITING MEMORY IN A TRANSFORMER
> [2] Knowledge Sanitization of Large Language Models

---

> > ### Author Response · Authors · 2024-11-19
> > **Official Rebuttal to Reviewer v4wY (Part 2/2)**
> >
> > **[Q1] Unlearning specific relations while retaining others:**
> > This is an interesting question, and we appreciate the opportunity to clarify. In our method, `O1` is only related to the pair `(S, R1)` and not to `(S, R2)`. When editing, we consider the full input context rather than simply editing all relations associated with `S`. Specifically, our method targets the relation `(S, R)`, meaning that editing `O1` as the outcome of `(S, R1)` does not affect `O2` as the outcome of `(S, R2)`.
> > This design ensures that our edits are precise and localized, enabling us to unlearn specific relations without unintentionally altering unrelated ones. We will explicitly include this explanation in the revised manuscript to better highlight the granularity and specificity of our approach.
> >
> >
> > **[Q2] Handling output behavior during unlearning:** We appreciate the reviewer’s concern regarding the output behavior of the model post-unlearning. To clarify, we do not aim to generate “wrong” answers. Instead, our approach disturbs the model’s distribution for the target knowledge, ensuring that the model “does not know” the correct answer. For instance, given the input “ChatGPT is developed by”, the model would no longer confidently output “OpenAI”. If the model remains fluent, it will likely generate some company name after “ChatGPT is developed by”, as this is a plausible continuation for a normal sentence. However, as the model inherently does not know the correct answer after unlearning, it would output a random or unrelated answer instead. This behavior aligns with the model’s state after erasing the specific knowledge.
> >
> > Potentially, this issue can be mitigated through instruction tuning, where the model could be encouraged to output “I don’t know” when it lacks the required knowledge. Intuitively, the model acquires knowledge during the pre-training stage, and our focus in this paper is to remove some knowledge from that model. As for outputting "I don't know", it might be more related to the model's behavior, which should be regularized during the instruction tuning stage. Thus, before applying instruction tuning, it is possibly reasonable to focus primarily on the model’s inherent knowledge without additional intervention.
> >
> > Our approach ensures the model no longer confidently outputs the erased knowledge, and it remains compatible with further refinement through instruction tuning. We will elaborate on this design choice and its implications in the revised paper.

---

### Official Review · Reviewer_xUFX · 2024-11-04

**Soundness:** 3
**Presentation:** 3
**Contribution:** 2
**Rating:** 6
**Confidence:** 4

**Summary:**

Large language models (LLMs) can memorize a vast amount of knowledge, which leads to concerns about the memorization of sensitive knowledge. This paper focuses on the problem of large-scale knowledge washing, that is, how to forget knowledge on a large scale while minimizing the impact on the model's reasonining ability. The authors propose a method named LAW, which is based on the model editing technology to update the MLP layers of the decoder model. It determines the keys and values related to the forgotten knowledge in a specific way, redefines the objective function, and considers practical factors. Experiments show that LAW performs excellently in both small-scale (zsRE and CounterFactual datasets) and large-scale (Wiki-Latest dataset) knowledge washing in terms of the cleanliness of knowledge forgetting and the maintenance of reasoning ability, outperforming many baseline methods.

**Strengths:**

1. The paper is well-written with clear logic.
2. The experiments consider both small- and large-scale knowledge washing settings and include baselines for both model editing and machine unlearning methods.
3. The ablation studies are thorough.

**Weaknesses:**

1. Model editing methods often face a problem of generalization where, after editing for a specific query, the model's response reverts to the pre-edit state when the query is rephrased. This raises the question of whether LAW truly makes the model forget sensitive knowledge or just forgets the specific case. It is necessary to use jailbreak prompts to verify true washing;
2. The abstract mentions that machine unlearning affects the fluency and reasoning ability of the model's generation. While reasoning ability is evaluated in subsequent experiments, the fluency of the generated text is not assessed.

**Questions:**

1. FT-UL is used, but there is no consideration of using FT-UL with utility loss as a baseline.
2. It would be better to conduct more experiments with the LLaMA or Qwen series.

---

> ### Author Response · Authors · 2024-11-18
> **Official Rebuttal to Reviewer xUFX (Part 1/2)**
>
> We sincerely thank the reviewers for the positive feedback and detailed comments! We appreciate it so much and would like to address the concerns as follows:
>
>
> **[W1] Validaty of Generalization**: We appreciate the reviewer’s concern regarding the generalization of knowledge unlearning. To address this, we conducted additional experiments using paraphrased queries from the zsRE and MCF datasets. This approach probes the model’s knowledge under varied rephrasings to evaluate whether LAW effectively unlearns knowledge beyond the specific case. The results are summarized in Table 1 below:
>
>
> | Model      | zsRE Acc   | zsRE QA-F1 | CounterFactual Acc | CounterFactual QA-F1 |
> |------------|------------|------------|--------------------|----------------------|
> | GPT2-XL    | 0.6790     | 0.1371     | 0.4632             | 0.1308               |
> | MEMIT      | 0.0710     | 0.0506     | 0.1958             | 0.1349               |
> | LAW        | **0.0140** | **0.0104** | **0.1274**         | **0.1027**           |
> | GPT-J-6B   | 0.6966     | 0.1537     | 0.5059             | 0.1507               |
> | MEMIT      | 0.0231     | 0.0102     | 0.2623             | 0.1037               |
> | LAW        | **0.0036** | **0.0026** | **0.1492**         | **0.0668**           |
>
>
> These results demonstrate that our method achieves robust generalization, as the model consistently forgets the target knowledge across paraphrased queries. This indicates that LAW does not merely remove specific instances but generalizes well to related prompts. We will include these findings in our paper to strengthen the discussion on the generalization capability of LAW.
>
>
> **[W2] Model Fluency:** We acknowledge the importance of assessing the fluency of the model’s generated text post-unlearning, as mentioned in the abstract. To address this, we evaluate the perplexity of the model on a validation set (we choose the first 500 examples from the snapshot `CC-MAIN-2024-10`) from Fineweb-Edu, a widely-used pre-training dataset. Perplexity serves as a proxy for fluency, as lower perplexity indicates more coherent and fluent text generation. The results are as follows: (As a reference, the perplexity of GPT2-XL and GPT-J-6B on the validation set is 2.66 and 2.31, repsectively)
>
> | Model          | zsRE  | CounterFactual | Wiki-Latest |
> |----------------|-------|----------------|-------------|
> | GPT2-XL-MEMIT  | 2.67  | 2.70           | 2.72        |
> | GPT2-XL-SeUL   | 2.69  | 10.82          | 96.95       |
> | GPT2-XL-LaW    | 2.68  | 2.73           | 2.75        |
> | GPT-J-6B-MEMIT | 2.31  | 2.33           | 2.37        |
> | GPT-J-6B-SeUL  | 2.34  | 2.33           | 121.94      |
> | GPT-J-6B-LaW   | 2.32  | 2.35           | 2.42        |
>
> These perplexity scores indicate that LAW minimally impacts the fluency of the model’s outputs, preserving its ability to generate coherent and high-quality text. On the contrary, unlearning methods with negative loss could affect the model’s fluency drastically during the case of large-scale washing. We will add these results and the corresponding discussion to the revised paper. Here we measure the generalization abilities using zsRE and CounterFactual as they have existing paraphrased questions in the dataset. For `Wiki-Latest`, we have to use some instruct model (like `gpt-4o-mini`) to rephrase the prompt and possibly need to do some post-processing to make sure the correctness of the paraphrased questions. We can certainly collect a subset of `Wiki-Latest` but that would lose the point of "Large-Scale". So we currently have not done it and we hope to expand the dataset and provide some paraphrased questions in our future work.

---

> > ### Author Response · Authors · 2024-11-18
> > **Official Rebuttal to Reviewer xUFX (Part 2/2)**
> >
> > **[Q1] FT-UL as a Baseline**: We would like to highlight that FT-UL refers to fine-tuning for unlearning (as stated in Line 298), which means we train the model using unlearning loss, i.e., the negative next-word-prediction loss, to forget the target knowledge. We will add clearer clarifications in the paper.
> >
> > **[Q2] Choices of models**: We understand the importance of expanding the scope of our experiments to include additional models like LLaMA or Qwen. However, our current implementation is built on top of the MEMIT repository, which provides effective implementations only for GPT-2 and GPT-J. While we attempted to adapt MEMIT for LLaMA (specifically llama2-7b), the efficacy score did not exceed 0.75, compared to 0.96+ for GPT-2 and GPT-J (shown in [1]). We assume this could result from configuration issues, such as selecting the appropriate layers to edit, tuning the value of $\lambda$ of Eq.(15) of  [1], or estimating the covariance matrix ($E_k[kk^T]$) using a more suitable dataset (it might need to be more aligned with LLaMA's pretraining set rather than the Wikitext used in [1], for this part, please see Appendix B.3 in [1]). Resolving these challenges would require significant adjustments unrelated to our proposed method. For these reasons, we chose to focus on GPT-2 and GPT-J in our experiments. We believe this decision allows us to present a fair and focused evaluation of LAW, while LLaMA integration remains a promising direction for future work. We will clarify this limitation in the revised manuscript.
> >
> > [1] MASS-EDITING MEMORY IN A TRANSFORMER

---

### Official Review · Reviewer_Y2iC · 2024-11-04

**Soundness:** 3
**Presentation:** 4
**Contribution:** 3
**Rating:** 6
**Confidence:** 3

**Summary:**

This paper presents LAW (Large Scale Washing), a novel method for removing a large set of factual knowledge from large language models (LLMs) while preserving reasoning abilities. The authors hypothesize that knowledge and reasoning abilities in LLMs are disentangled. LAW is inspired by model editing, which focuses on adding factual relations, while LAW deletes factual relations. Experiments on datasets (zaRE and CounterFactual), also including a new large-scale dataset called Wiki-Latest, demonstrate that LAW achieves better knowledge removal than other methods while maintaining the model's reasoning abilities.

**Strengths:**

Novelty: The paper proposes a novel objective function specifically designed to remove knowledge represented in triplet format, achieving an approach distinct from existing methods.

Creation of a large-scale dataset: The development of Wiki-Latest, a new large-scale dataset derived from Wikipedia triplets, is a valuable contribution to the field.

Comprehensive evaluation: The paper presents a detailed comparative analysis with multiple existing methods.

**Weaknesses:**

Knowledge vs. Reasoning: While the paper aims to disentangle knowledge and reasoning, it doesn't explicitly define what constitutes "reasoning," while the knowledge is defined as triples. Are they referring to specific modules or functionalities within the model? Or are they more abstract concepts related to the model's behavior?

Insufficient discussion on disentanglement: While the paper claims in section 5.2 that "In this paper, we show the possibility of the disentanglement between knowledge and reasoning by washing a large amount of knowledge from the model while only minimally affecting the reasoning abilities," this assertion doesn't seem to be sufficiently discussed in the later sections. It lacks a theoretical foundation to support the claim that knowledge and reasoning are truly disentangled in the model. The paper primarily relies on empirical evidence from the experiments, and it could be strengthened by exploring the underlying mechanisms that allow for this separation.

The limited scope of knowledge washing: They lack the ability to address the washing of knowledge expressed in more general or abstract forms, such as knowledge embedded within the text that doesn't have a clear subject-relation-object structure. It might be beneficial to consider and compare an abstract editing method that modifies behavior at the sentence level, such as DINM, proposed in 'Detoxifying Large Language Models via Knowledge Editing' (ACL 2024) (https://aclanthology.org/2024.acl-long.171/).

**Questions:**

The choice of reasoning benchmark: It is not clear why the three benchmark datasets (LAMBADA, HellaSwag, ARC_easy) were chosen from many other reasoning benchmark datasets. The setup for LAMBADA and HellaSwag is word/sentence prediction, which does not seem to be the "reasoning" ability that this paper tries to disentangle.  Can something more like logical and/or numerical reasoning, such as GSM8K, be added?

Validity of  `<|endoftext|>` output: Could you provide a more detailed explanation of the rationale behind configuring the model to output `<|endoftext|>` in place of the original object when deleting knowledge? Are there any existing research studies that employ a similar task setup? If this configuration is specific to this study, please clearly state so. Additionally, could you explain the fundamental difference between outputting `<|endoftext|>` (deletion) and outputting another appropriate object (editing)?

Model architecture clarification: Although equation (1) in Section 3 illustrates a parallel model structure with Attention and MLP (like GPT-J), the experiments also include GPT2-XL, which has a serial structure. To prevent misunderstanding, it might be beneficial to explicitly state that the proposed method is not dependent on any specific model architecture.

---

> ### Author Response · Authors · 2024-11-18
> **Official Rebuttal to Reviewer Y2iC (Part 1/2)**
>
> We sincerely thank the reviewer for the detailed comments and suggestions! We address the concerns as below:
>
> **[W1] Knowledge vs. Reasoning**: By “reasoning,” we refer to tasks that do not require the model to have prior domain knowledge, such as context-based question answering, mathematical reasoning, etc. These are more abstracted concepts about the dataset property rather than the model’s specific modules or functionalities.  We will clarify this definition further in our paper, explaining that knowledge and reasoning are treated as distinct task types for our evaluations.
>
> **[W2] Insufficient discussion on disentanglement**: Our paper empirically shows the possibilities of knowledge-reasoning disentanglement by washing a large amount of knowledge without affecting reasoning abilities. We will add more discussions in the later sections (the experiments part and the conclusion part). As for theoretical validation, our current understanding is that the attention layers might not be as critical for encapsulating knowledge or reasoning, with MLP layers playing a more significant role. Given that MLP layers can be decomposed into keys and values, it is conceivable that some keys and values pertain to reasoning ability, while others relate to knowledge, and some might be relevant to both. Exploring the hypothesis that certain keys and values are associated with reasoning ability could serve as a theoretical basis for disentangling knowledge and reasoning. However, we acknowledge that identifying these specific keys and values is a complex task that warrants further investigation. We consider this an intriguing avenue for future research and thank the reviewer for highlighting its importance.
>
> **[W3] The limited scope of knowledge washing**: We acknowledge that our method currently focuses on triplet-structured knowledge, which may seem limited. However, triplet formats are widely accepted as a structured form of factual knowledge. We are keen to explore extending LaW to handle unstructured knowledge in future work, as mentioned in our conclusion. We thank the reviewer for pointing out DINM, which provides a relevant perspective on free-form knowledge editing, and we will incorporate a discussion of this approach into our paper to broaden the scope and consider DINM as a baseline after extending LaW into unstructured knowledge washing.

---

> ### Author Response · Authors · 2024-11-18
> **Official Rebuttal to Reviewer Y2iC (Part 2/2)**
>
> **[Q1] The choice of reasoning benchmark**: The benchmarks we selected (LAMBADA, HellaSwag, ARC_easy) are widely used in the `lm-eval` library, ensuring that models like GPT2-XL and GPT-J-6B achieve meaningful scores. This makes them suitable for our comparison as they represent various reasoning tasks in an accessible way for our chosen models. However, we acknowledge that logical and numerical reasoning benchmarks, such as GSM8K, could further clarify reasoning evaluation. We include the results on GSM-8K as shown in the table below:
>
> | Model                        | flexible-extract        | strict-match          |
> |------------------------------|-------------------------|-----------------------|
> | GPT2-XL                      | 0.0205 ± 0.0039        | 0.0121 ± 0.0030      |
> | GPT2-XL-MEMIT-MCF            | 0.0182 ± 0.0037        | 0.0099 ± 0.0027      |
> | GPT2-XL-LaW-MCF              | 0.0197 ± 0.0038        | 0.0121 ± 0.0030      |
> | GPT2-XL-MEMIT-zsRE           | 0.0190 ± 0.0038        | 0.0083 ± 0.0025      |
> | GPT2-XL-LaW-zsRE             | 0.0205 ± 0.0039        | 0.0106 ± 0.0028      |
> | GPT2-XL-MEMIT-Wiki           | 0.0167 ± 0.0035        | 0.0083 ± 0.0025      |
> | GPT2-XL-LaW-Wiki             | 0.0212 ± 0.0040        | 0.0099 ± 0.0027      |
> | GPT-J-6B                     | 0.0425 ± 0.0056        | 0.0349 ± 0.0051      |
> | GPT-J-6B-MEMIT-MCF           | 0.0440 ± 0.0056        | 0.0364 ± 0.0052      |
> | GPT-J-6B-LaW-MCF             | 0.0394 ± 0.0054        | 0.0334 ± 0.0049      |
> | GPT-J-6B-MEMIT-zsRE          | 0.0394 ± 0.0054        | 0.0356 ± 0.0051      |
> | GPT-J-6B-LaW-zsRE            | 0.0376 ± 0.0051        | 0.0336 ± 0.0049      |
> | GPT-J-6B-MEMIT-Wiki          | 0.0440 ± 0.0056        | 0.0379 ± 0.0053      |
> | GPT-J-6B-LaW-Wiki (x)        | 0.0379 ± 0.0051        | 0.0326 ± 0.0047      |
>
> From these results, we can see that this result is consistent with the results in the paper: our model can minimally affect the model’s reasoning abilities and the effects are comparable with using MEMIT (though ours can wash knowledge more thoroughly.)
>
> **[Q2] The choice of using `<|endoftext|>` as the new output**: Consider a factual statement “ChatGPT is developed by OpenAI”, we also would like to mention that the appropriate object in this example is `OpenAI` and there is no other objects here. This leads to a different setting as model editing.
> - As there is no other object here, we need to add a new object to formulate the unlearning problem into model-editing problem. We are the first to propose using `<|endoftext|>` as the new object, which serves as a novel approach to indicate “deletion”. To the best of our knowledge, none has directly applied model-editing methods in unlearning tasks thus we need to propose some adaptation ourselves. Using `<|endoftext|>` serves as a potential proxy for forgetting, as the model no longer produces content after the input.
> - Using `<|endoftext|>` means that we are editing this factual statement into “ChatGPT is developed by <|endoftext|>”, while LaW proposes to forget the relation between ChatGPT and OpenAI so the distribution over the vocabulary after “ChatGPT is developed by” is disturbed. Fundamentally, LaW propose to delete the relation between ChatGPT and OpenAI but using `<|endoftext|>` is editing the relation. Thus we argue our proposal LaW intuitively makes more sense and it also shows better performances in our experiments.
>
> **[Q3] Clarification of Model Architectures**: Thank you for raising this point! We will clarify that our method is model-agnostic and applies to any transformer-based model with MLP layers, regardless of architecture specifics such as parallel or serial attention-MLP configurations. This will be added in Section 5 to prevent any potential confusion.

---

> > ### Comment · Reviewer_Y2iC · 2024-11-22
> >
> > (Thank you for your clarification! I updated my score accordingly.)

---

> ### Author Response · Authors · 2024-11-22
> **Response to Reviewer Y2iC**
>
> Dear Reviewer Y2iC,
>
> We are glad that we have addressed your concerns, thank you so much!
>
> Best,
> Authors

---

### Meta-Review · Area_Chair_RGYj · 2024-12-20

**Metareview:**

This paper introduces LAW (Large Scale Washing), a novel method for removing a large set of factual knowledge from large language models (LLMs) while preserving reasoning abilities. The authors hypothesize that knowledge and reasoning abilities in LLMs are disentangled. LAW is inspired by model editing, which focuses on adding factual relations, while LAW deletes factual relations. Experiments on datasets (zaRE and CounterFactual), along with a new large-scale dataset called Wiki-Latest, demonstrate that LAW outperforms other methods in knowledge removal while maintaining the model's reasoning abilities.  The authors need to carefully revise the paper based on the reviewers' comments, add discussions and comparisons of related work, include additional experiments, and further improve the paper.

**Additional Comments On Reviewer Discussion:**

The reviewers have reached a consensus in their discussion.

---

### Decision · Program_Chairs · 2025-01-22

Accept (Poster)